# The mRNA methyltransferase Mettl3 modulates cytokine mRNA stability and limits functional responses in mast cells

Cristina Leoni [1,4] ✉, Marian Bataclan [1,4], Taku Ito-Kureha [2], Vigo Heissmeyer [2,3] & Silvia Monticelli [1] ✉

Mast cells are central players in allergy and asthma, and their dysregulated responses lead to reduced quality of life and life-threatening conditions such as anaphylaxis. The RNA modification $N^6$-methyladenosine ($m^6A$) has a prominent impact on immune cell functions, but its role in mast cells remains unexplored. Here, by optimizing tools to genetically manipulate primary mast cells, we reveal that the $m^6A$ mRNA methyltransferase complex modulates mast cell proliferation and survival. Depletion of the catalytic component Mettl3 exacerbates effector functions in response to IgE and antigen complexes, both in vitro and in vivo. Mechanistically, deletion of Mettl3 or Mettl14, another component of the methyltransferase complex, lead to the enhanced expression of inflammatory cytokines. By focusing on one of the most affected mRNAs, namely the one encoding the cytokine IL-13, we find that it is methylated in activated mast cells, and that Mettl3 affects its transcript stability in an enzymatic activity-dependent manner, requiring consensus $m^6A$ sites in the *Il13* 3'-untranslated region. Overall, we reveal that the $m^6A$ machinery is essential in mast cells to sustain growth and to restrain inflammatory responses.

Mast cells are tissue-resident immune cells whose functional impact in the context of immune responses besides allergy and asthma is still incompletely understood[1]. They respond very rapidly to a large variety of stimuli with the production of immune- and vaso-active mediators such as cytokines, histamine and prostaglandins. Cytokine release by mast cells can have immunoregulatory functions in both health and disease. For instance, apart from the established role for the cytokine IL-13 in allergic inflammation and airway mucus production, the directional secretion of TNF by perivascular mast cells was shown to induce the recruitment of neutrophils during skin inflammation[2].

The appropriate control of mast cell differentiation and functions requires a network of transcriptional and post-transcriptional mechanisms of regulation of gene expression that allows fast

responses to danger signals while limiting excessive damage to the host. How mast cell responses are regulated is still a matter of active investigation. For example, our lab found that chromatin dynamics and epigenetic modifications such as 5-methylcytosine and 5-hydroxymethylcytosine in the genomic DNA are central regulators of mast cell responses and proliferation, respectively[3–5]. However, what is the role of post-transcriptional modifications in modulating mast cell responses is still unknown.

The RNA modification $N^6$-methyladenosine ($m^6A$) has important regulatory functions in gene expression. In the context of messenger RNA (mRNA), $m^6A$ is deposited co-transcriptionally by the action of a multi-subunit methyltransferase complex composed of the enzymatic component Mettl3 and other accessory proteins, such as Mettl14 and

[1]Institute for Research in Biomedicine, Università della Svizzera italiana (USI), 6500 Bellinzona, Switzerland. [2]Institute for Immunology, Biomedical Center, Faculty of Medicine, Ludwig-Maximilians-Universität in Munich, 82152 Planegg-Martinsried, Germany. [3]Research Unit Molecular Immune Regulation, Helmholtz Zentrum München, 81377 Munich, Germany. [4]These authors contributed equally: Cristina Leoni, Marian Bataclan. ✉e-mail: cristina.leoni@irb.usi.ch; silvia.monticelli@irb.usi.ch

Wtap (Wilms tumor 1-associating protein), which are also essential for m⁶A deposition[6]. The functional outcome of m⁶A is mediated by its recognition by RNA-binding proteins primarily of the YTH family, which upon binding frequently affect mRNA stability[7,8]. Since it influences basic aspects of post-transcriptional gene regulation, the m⁶A modification also controls various aspects of innate and adaptive immunity, including immune recognition of invading pathogens, cell activation and differentiation[9]. For example, deletion of *Mettl3* or *Mettl14* in hematopoietic stem cells affected their output in terms of maintenance of cell identity and differentiation[9–11]. In the myeloid compartment, deletion of *Mettl3* in dendritic cells resulted in impaired maturation of these cells in response to lipopolysaccharide (LPS), leading to their reduced ability to stimulate T cell responses[12]. In macrophages, deletion of *Mettl3* limited the expression of inflammatory cytokines in response to LPS and impaired effector responses, leading to increased susceptibility to bacterial infection and uncontrolled tumor growth[13,14]. However, this phenotype appeared to be at least in part dependent on the specificities of the experimental system, since another study found improved immunosuppression of tumor-infiltrating myeloid cells lacking *Mettl3*[15], while targeting Mettl3 using small molecule inhibitors had a strong anti-tumor activity in acute myeloid leukemia, leading to increased cell differentiation and apoptosis[16]. Finally, ablation of *Mettl14* led to unrestrained responses of macrophages to bacterial infection, resulting in sustained expression of inflammatory cytokines and death[17]. Overall, while these studies highlight a crucial importance of the methyltransferase complex and of the m⁶A modification in the myeloid compartment, the mechanistic aspects remain incompletely understood.

In mast cells, the role of m⁶A in regulating cell differentiation and functions is unknown[18]. To address this question, we first optimized CRISPR-Cas9 tools to effectively perform gene editing in mast cells, allowing us to dissect the role of the m⁶A methyltransferase complex in mature mast cells, without affecting cell development. Then, we found that mast cells lacking Mettl3 developed exacerbated effector functions in response to stimulation with IgE and antigen complexes, both in vitro and in vivo. Mettl3 deficiency induced increased stability of inflammatory transcripts, most notably of the *Il13* mRNA, and IL-13 expression was diminished upon overexpression of Mettl3. Such effect depended both on the Mettl3 enzymatic activity and on specific adenosines in the 3'-untranslated region (3'UTR) of the *Il13* transcript. Overall, our findings point towards a direct link between Mettl3 expression, its enzymatic activity and m⁶A deposition in the regulation of the stability of inflammatory transcripts in mast cells, as well as in sustaining their basal proliferation.

## Results

### Mettl3 regulates mast cell proliferation and effector functions

We measured the expression of key components of the m⁶A methyltransferase complex in response to stimulation of murine mast cells. Activation with IgE and antigen complexes or PMA and ionomycin induced Mettl3 expression, both at mRNA and protein level (Fig. 1a and Supplementary Fig. 1a, b). An increase in the expression of Mettl3 was also observed upon stimulation of peritoneal-derived mast cells (PMCs) (Fig. 1b and Supplementary Fig. 1c). Since Mettl3 is the catalytic component of the m⁶A methyltransferase complex, we set out to investigate its role in acute mast cell responses.

First, we depleted Mettl3 in mast cells by RNA interference (RNAi). Transient transfection of differentiated mast cells with small-interfering RNAs (siRNAs) against *Mettl3* was >95% efficient (Supplementary Fig. 1d) and led to the partial depletion of *Mettl3* mRNA and protein after 48 h (Fig. 1c and Supplementary Fig. 1e). Upon stimulation with IgE and antigen complexes, mast cells depleted of Mettl3 showed significantly increased ability to produce the inflammatory cytokines IL-6, TNF and IL-13, measured by intracellular staining (Fig. 1d). Unstimulated mast cells did not express detectable cytokines

(Supplementary Fig. 1f, gating strategies are shown in Supplementary Fig. 1g). Overexpression of Mettl3 by lentiviral transduction led to the opposite effect, namely reduced cytokine expression (Fig. 1e and Supplementary Fig. 2a, b), suggesting a role in regulating mast cell effector functions. Overexpression of Mettl3 also modestly enhanced in vitro mast cell differentiation (Supplementary Fig. 2c). Upon depletion of *Mettl3*, mast cells released the content of pre-stored cytoplasmic granules at a greater extent, further suggesting overall hyper-activation in response to IgE stimulation (Fig. 1f). However, reducing *Mettl3* expression also significantly impaired the ability of the cells to proliferate, measured by BrdU incorporation (Fig. 1g). Comparable results were obtained upon deletion of human METTL3 from the human mast cell lines HMC-1.1 and 1.2 (Supplementary Fig. 2d), indicating that the pathways regulated by m⁶A in mast cells are likely to be conserved. These data point towards an important role of Mettl3 in regulating mast cell functions, leading to enhanced effector functions but reduced proliferation capacity.

### Mettl3 restrains mast cell responses in vivo

To further dissect the role of Mettl3 in mast cells we optimized a CRISPR-Cas9 transfection protocol to efficiently delete genes of interest in primary mast cells. As a proof of principle, we first optimized this approach by ablating the expression of the surface receptor c-Kit, since it is highly expressed by mast cells, and there was no addition of SCF (the c-Kit ligand) in these in vitro cultures. Transfection of Cas9 ribonucleoproteins (RNPs) containing gRNAs directed against the *Kit* gene led reproducibly to high efficiency of transfection and deletion of the receptor in >90% of the cells (Fig. 2a and Supplementary Fig. 3a). Next, using two different gRNAs directed against the *Mettl3* gene, we achieved >60% deletion efficiency, as shown by intracellular staining for the Mettl3 protein (Fig. 2b). T7 endonuclease I assay on cells transfected with one or two gRNAs against *Mettl3* also showed the expected pattern of digestion (Fig. 2c and Supplementary Fig. 3b), demonstrating highly efficient CRISPR-Cas9-mediated deletion of genes of interest in mast cells. Deletion of the Mettl3 protein in more than half of the population did not impact the surface expression of c-Kit and FcεRIα (Supplementary Fig. 3c), and it was already sufficient to detect a significant physiologic response, as shown by the increased production of inflammatory cytokines (Fig. 2d, e) and by the reduction in cell proliferation (Fig. 2f), that was consistent with the phenotypes measured in cells in which Mettl3 expression was depleted by RNAi. Mettl3-deleted cells also showed reduced viability (Fig. 2g). Deletion of Mettl3 in mast cells differentiated in the presence of IL-3 + SCF (which induces a more mature phenotype) led to a similar increase in cytokine production and cell death, although the presence of SCF enabled the cells to bypass the defect in proliferation (Supplementary Fig. 3d–f). Mast cells lacking Mettl3 showed increased production of inflammatory cytokines also in response to IgE cross-linking in combination with IL-33 stimulation (Supplementary Fig. 3g). Next, we investigated the in vivo effects of *Mettl3* deletion in mast cells using a model of passive cutaneous anaphylaxis (PCA). Mast cells transfected with Cas9 RNPs against Mettl3 were injected intradermally into the ear pinna of c-Kit^{W-sh/W-sh} mice, that do not have mast cells[19]. After repopulation of the tissues, IgE anti-DNP antibodies were injected intradermally followed, 24 h later, by intravenous injection of HSA-DNP antigen and Evan's blue dye. The extent of dye extravasation induced by mast cell activation was measured by extracting the dye from the tissue and by measuring the $OD_{600}$ of the solution. We found that the extent of PCA was significantly increased in the absence of Mettl3 in vivo, underlining a role for m⁶A-mediated processes in restraining mast cell responses to IgE activation (Fig. 2h).

### Deletion of Wtap and Mettl14 phenocopy Mettl3 ablation

Although Mettl3 is the catalytic subunit of the methyltransferase complex, deletion of the accessory protein Wtap also abrogates m⁶A

methylation[6,20]. We therefore investigated the impact of deleting Wtap in differentiated mast cells. We took advantage of mice with a *Wtap^{fl/fl}* conditional allele[20]. We transduced bone marrow cells with a lentivirus expressing a GFP-Cre fusion protein to delete *Wtap* and we measured mast cell activation and proliferation. First, we found that deletion of the *Wtap* gene was efficient, as shown by intracellular staining for the Wtap protein (Fig. 3a). However, deletion of Wtap destabilized Mettl3 expression, that was reduced in *Wtap*-deleted mast cells (Fig. 3b). Consistent with the phenotype observed upon depletion of Mettl3, cells lacking Wtap showed strongly reduced proliferation (Fig. 3c), possibly also linked to the reduced expression of Mettl3. Surprisingly, surface staining to assess the expression of the mast cell markers c-Kit and FcεRIα revealed a strong and significant downregulation of the latter (Fig. 3d), an effect that was not observed upon depletion or

deletion of Mettl3 (Supplementary Fig. 3c). This observation points towards some m6A-independent effects of Wtap, possibly on mRNA splicing[21,22] and precluded the detailed analysis of IgE responses, due to the confounding effects linked to the downregulation of the IgE receptor.

Mettl3 exists as a heterodimer with Mettl14[23,24]. We deleted Mettl14 in mast cells by CRISPR-Cas9 (scheme in Fig. 4a) and we found that, similarly to the Mettl3 knock-out, deletion of Mettl14 led to decreased proliferation and viability (Fig. 4b, c), and increased cytokine expression (Fig. 4d) in mast cells. We also found that deletion of Mettl14 led to reduced Mettl3 expression (Fig. 4e, f). Similarly, deletion of Mettl3 led to reduced Mettl14 expression, while Wtap levels were increased (Supplementary Fig. 4). Virma expression was also modestly increased, albeit not significantly. These results suggest that the m6A

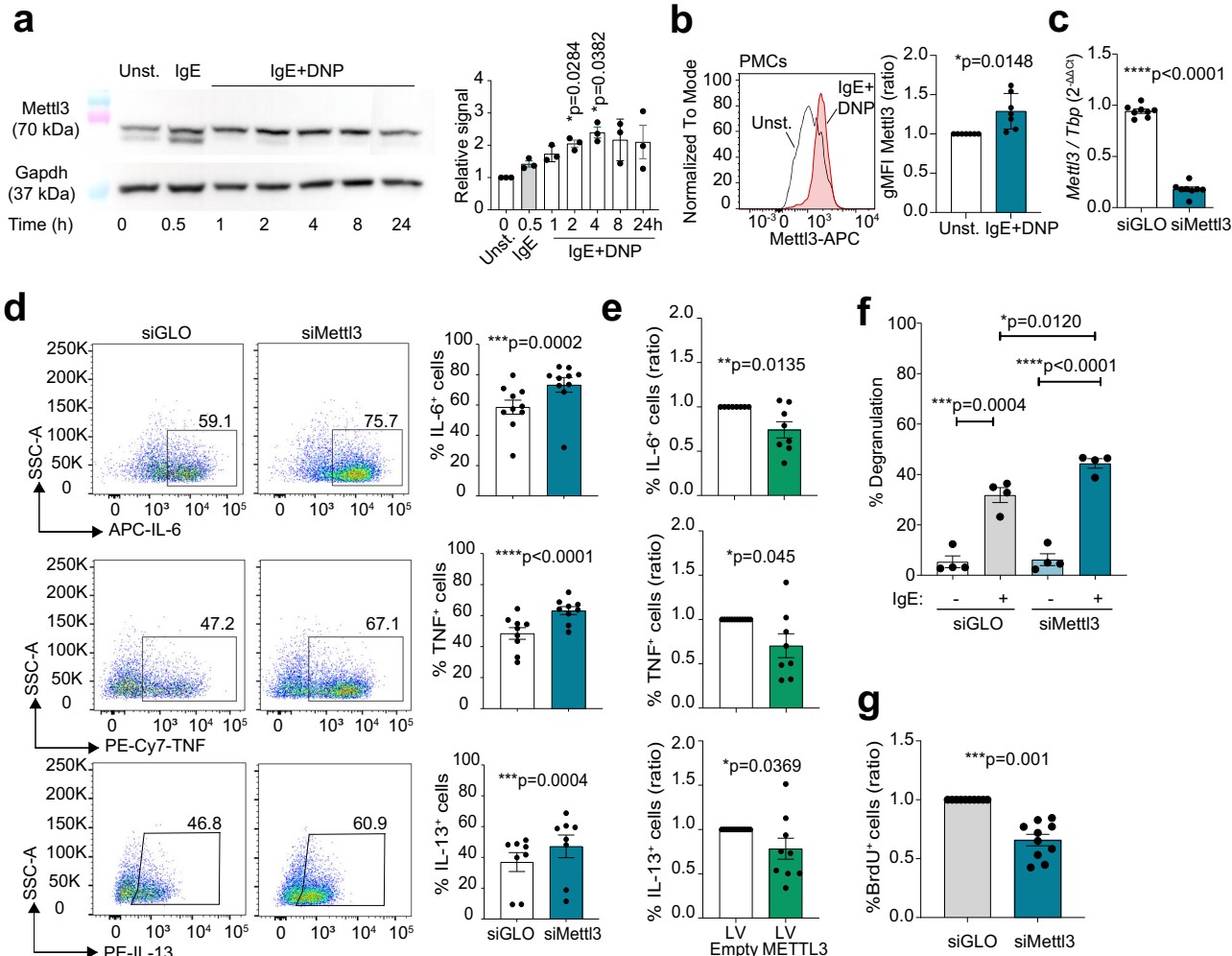

**Fig. 1 | Mettl3 modulates mast cell proliferation and effector functions. a** Mast cells were stimulated with IgE with or without HSA-DNP for the indicated times and expression of Mettl3 was analyzed by western blot. Left, one representative blot. Right, quantification of $N = 3$ independent experiments. Mean ± SEM. One-way ANOVA. **b** After ex-vivo expansion, PMCs were stimulated with IgE and antigen complexes and Mettl3 expression was measured by intracellular staining. $N = 7$ independent experiments. Mean ± SD. Paired $t$ test, two-tailed. **c** Expression of *Mettl3*, measured by RT-qPCR, in mast cells transfected with siRNAs against *Mettl3* or a non-targeting control. $N = 8$ independent experiments. Mean ± SEM. Paired $t$ test, two-tailed. **d** Expression of inflammatory cytokines measured by intracellular cytokine staining in mast cells transfected with siRNAs against *Mettl3* or a non-targeting control. Cells were stimulated with IgE and antigen complexes for 4 h. Left, representative FACS plots. Right, quantification of $N = 8$-10 independent

experiments. Mean ± SEM. Paired $t$ test, two-tailed. **e** Mast cells were transduced with lentiviral vectors to overexpress Mettl3. After selection of the transduced cells and stimulation with IgE and antigen complexes for 4 h, intracellular cytokine staining was performed to measure the expression of inflammatory cytokines. $N = 8$ independent experiments (ratio compared to controls). Mean ± SEM. Unpaired $t$ test, two-tailed. **f** Mast cells were transfected with siRNAs against *Mettl3* or a non-targeting control. 48 h after transfection, cells were stimulated with IgE and antigen complexes for 4 h, followed by β-hexosaminidase assay to measure the extent of degranulation. $N = 4$ independent experiments. Mean ± SEM. Unpaired $t$ test, two-tailed. **g** Mast cells were transfected with siRNAs against *Mettl3* or a non-targeting control, followed by BrdU incorporation to measure cell proliferation. $N = 10$ independent experiments. Mean ± SEM. Paired $t$ test, two-tailed. Source data are provided as a Source data File.

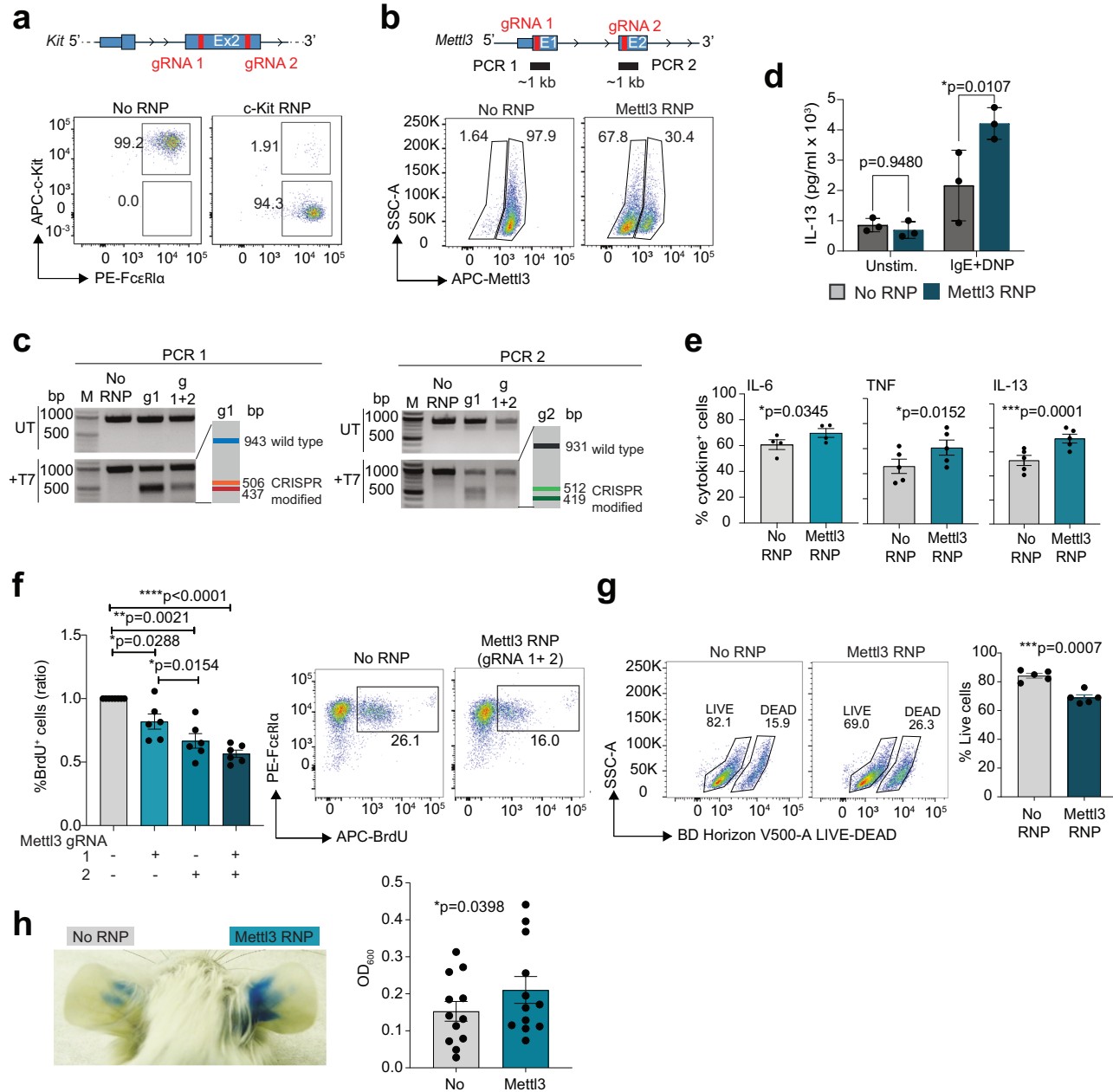

**Fig. 2 | Deletion of *Mettl3* by CRISPR-Cas9 alters mast cell responses.**
**a** Optimization of gene editing in primary mast cells. Cells were transfected with Cas9 RNPs containing gRNAs against the *Kit* gene (schematic representation on top). One representative FACS plot showing the efficiency of deletion of the surface receptor c-Kit. Representative of at least *N* = 8 experiments. **b** Top, schematic representation of the *Mettl3* locus, with indicated the location of the gRNAs and the PCRs used in T7 endonuclease I assays. Bottom: one representative FACS plot showing the deletion of the Mettl3 protein, by intracellular staining. Representative of at least *N* = 11 experiments. **c** T7 endonuclease I assay performed on genomic DNA from cells as in (**b**). The expected patterns of digestion, based on the locations of the sgRNAs and the PCR primers, are shown schematically on a side (see Supplementary Fig. 3b for further details). The effect of one gRNA or two gRNAs is shown. UT: untreated (no T7 enzyme). Representative of *N* = 4 experiments; M=marker. **d** Mast cells were transfected as in (**c**), followed by stimulation with IgE and antigen complexes for 6 h. Release of IL-13 in the culture supernatants was measured by ELISA. *N* = 3 independent experiments. Mean ± SD. Two-ways ANOVA. **e** Intracellular cytokine staining was performed on cells as in (**c**). *N* = 4-5 independent experiments. Mean ± SEM. Paired *t* test, two-tailed. **f** BrdU incorporation assay to measure proliferation 4-6 days post-transfection. *N* = 6 independent experiments (ratio compared to control samples). Mean ± SEM. Paired *t* test, two-tailed. One representative FACS plot is shown on the right. **g** Viability of mast cells transfected with Mettl3 RNPs or controls was measured by LIVE/DEAD staining. *N* = 5 independent experiments. Mean ± SEM. Paired *t* test, two-tailed. **h** Mast cells were injected into the ear pinna of Kit^W-sh/W-sh^ mice. Two weeks after, anti-DNP IgE antibodies were injected intradermally followed, 24 h later, by intravenous injection of HSA-DNP antigen together with Evan's blue dye. The dye was then extracted and measured spectrophotometrically. Each dot represents one mouse (*N* = 12), *N* = 2 independent experiments. One representative experiment is shown on the left. Mean ± SD. Paired *t* test, two-tailed. Source data are provided as a Source data File.

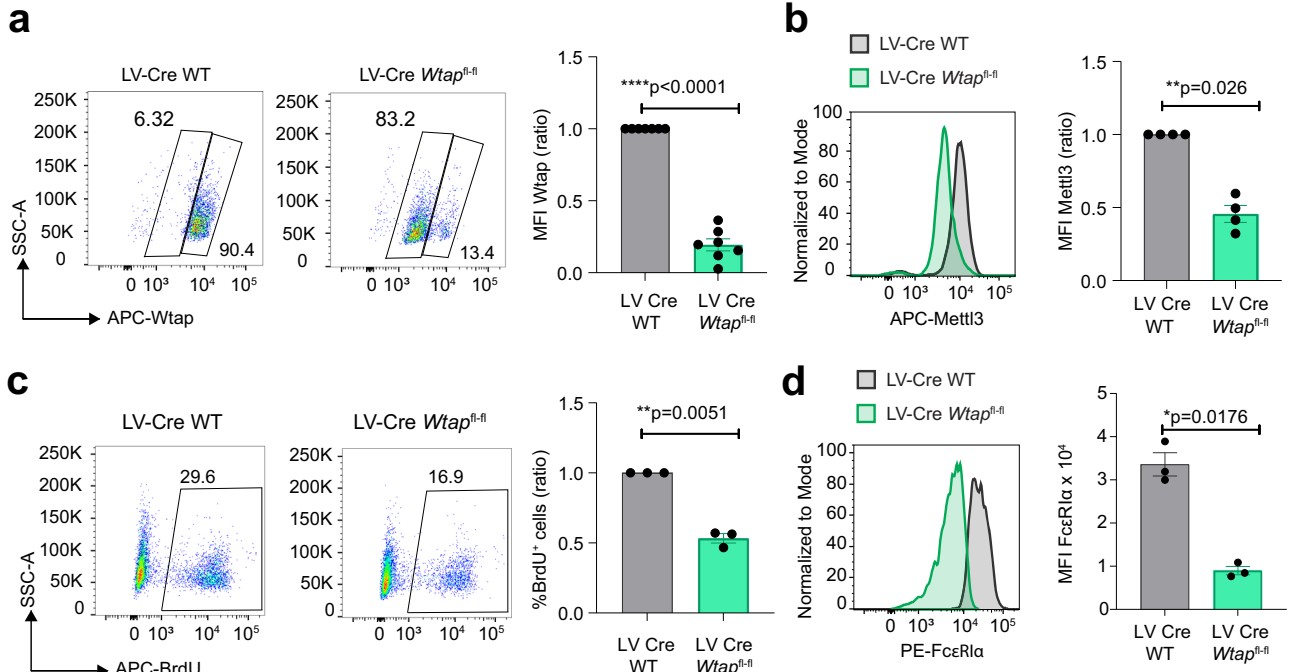

**Fig. 3 | Wtap deficiency impairs mast cell proliferation and FcεRI expression.**
**a** *Wtap^(fl/fl)* bone marrow cells were transduced with a lentiviral vector expressing a
GFP-Cre fusion protein, following by sorting of the GFP⁺ cells and intracellular
staining to verify Wtap protein depletion. Left, one representative staining for Wtap
expression. Right, Wtap mean fluorescence intensity (MFI) in *N* = 7 independent
experiments (ratio compared to control samples). Mean ± SD. Paired *t* test, two-
tailed. **b** Expression of Mettl3 measured by intracellular staining in cells as in (**a**).
*N* = 4 independent experiments. Mean ± SD. Paired *t* test, two-tailed. **c** Proliferation
(measured by BrdU incorporation) of cells as in (**a**). Left, one representative
experiment. Right, results of *N* = 3 independent experiments (ratio compared to
control samples). Mean ± SEM. Paired *t* test, two-tailed. **d** Expression of surface
FcεRI in cells lacking Wtap. Left, one representative experiment. Right, results of
*N* = 3 independent experiments. Mean ± SEM. Paired *t* test, two-tailed. Source data
are provided as a Source data File.

machine complex in its entirety is required to maintain mast cell
homeostasis and response to stimuli, and that removal of any of its
components affects the stoichiometry of the entire complex in an
unpredictable manner.

## Inflammatory transcripts are m⁶A-methylated in stimulated mast cells

To dissect the transcriptome changes that occur in mast cells depleted
for Mettl3 we first performed RNA-seq on cells transfected with Mettl3
RNPs (Fig. 5a and Supplementary Data 1). We found that at resting state
214 genes were downregulated and 487 were significantly upregulated
in absence of Mettl3 (*p* value≤0.05 and fold change ≥1.5). Down-
regulated transcripts included *Mettl3* itself and other regulators of
mast cell activation and function, such as *Hck*, encoding a Src family
kinase involved in the regulation of mast cell activation and cytoske-
letal reorganization, and described to positively regulate the pro-
liferation of mast cells[25]. Other prominently downregulated genes
included *Cpe*, encoding a carboxypeptidase enzyme, and *Il2ra*, that
was already described to be associated to altered mast cell phenotypes
in the mouse[26]. Several of the upregulated genes instead encoded
regulators of cell proliferation and intracellular trafficking, including
the tumor suppressor *Tspan32* and the actin-binding protein *Parvb*,
concordant with the phenotypes observed in mast cells lacking Mettl3.
Basal levels of expression of the *Tnf* transcript were also increased.
Indeed, gene ontology analysis revealed only few significant categories
for the downregulated genes, while the upregulated genes were
included in categories associated to the cytoskeleton (cell shape,
migration, adhesion, intracellular trafficking) and also apoptosis and
cell proliferation (Fig. 5b).

Although some of the dysregulated mRNAs may be direct targets
of Mettl3 catalytic activity, the expression of other transcripts may be
affected in an indirect manner. To determine which transcripts were

specifically methylated in mast cells before or after stimulation, we
performed m⁶A-crosslinking, immunoprecipitation and sequencing
(miCLIP-seq), which allows the detection of m⁶A at near-nucleotide
resolution[27]. The m⁶A enrichment at different parts of the mRNA of
all targets, showed corresponding signals throughout the mRNA, and
enrichment at the boundary between the coding sequence and the
3'UTR (Fig. 5c). A prominent methylation signal was also observed
within the 5'UTR in both resting and activated cells, consistent with
the fact that miCLIP-seq also identifies m⁶Am modifications occur-
ring at the first nucleotide of transcripts[27], which may suggest addi-
tional roles in the process of translation. Despite many transcripts
being methylated in resting mast cells, there was limited overlap (183
transcripts) with genes differentially expressed upon Mettl3 deple-
tion (Fig. 5d). Within these 183 transcripts, the majority (137) were
upregulated upon Mettl3 depletion (Fig. 5e), suggesting that they
may include transcripts directly affected in their half-life by the m⁶A
modification. These included the transcripts encoding the Rho-
related GTP-binding proteins *Rhob, Rhog* and *Rhoh*, which are reg-
ulators of a wide variety of cellular processes including actin cytos-
keleton reorganization, the *Inf2* gene, involved in actin
polymerization and depolymerization, as well as transcriptional
regulators (*Egr1, Rxra, Sox12*), metabolic enzymes (*Chst12, Naglu*) and
regulators of nuclear-cytoplasmic shuttling (*Nxt1*). Other genes were
found to be methylated, but they were not significantly affected by
Mettl3 deletion in mast cells. These included the mRNAs encoding
the negative regulators of cytokine signaling *Socs1* and *Socs3* (that
were found to be increased in T lymphocytes lacking *Mettl3*[28]), sug-
gesting that m⁶A methylation may not be the predominant
mechanism of regulation of these transcripts in mast cells. Finally,
upon acute stimulation, additional transcripts emerged that were
methylated in mast cells, including many inflammatory transcripts
such as *Il13, Csf1, Tnf, Il6, Csf2, Il3* (Fig. 5f and Supplementary Data 2).

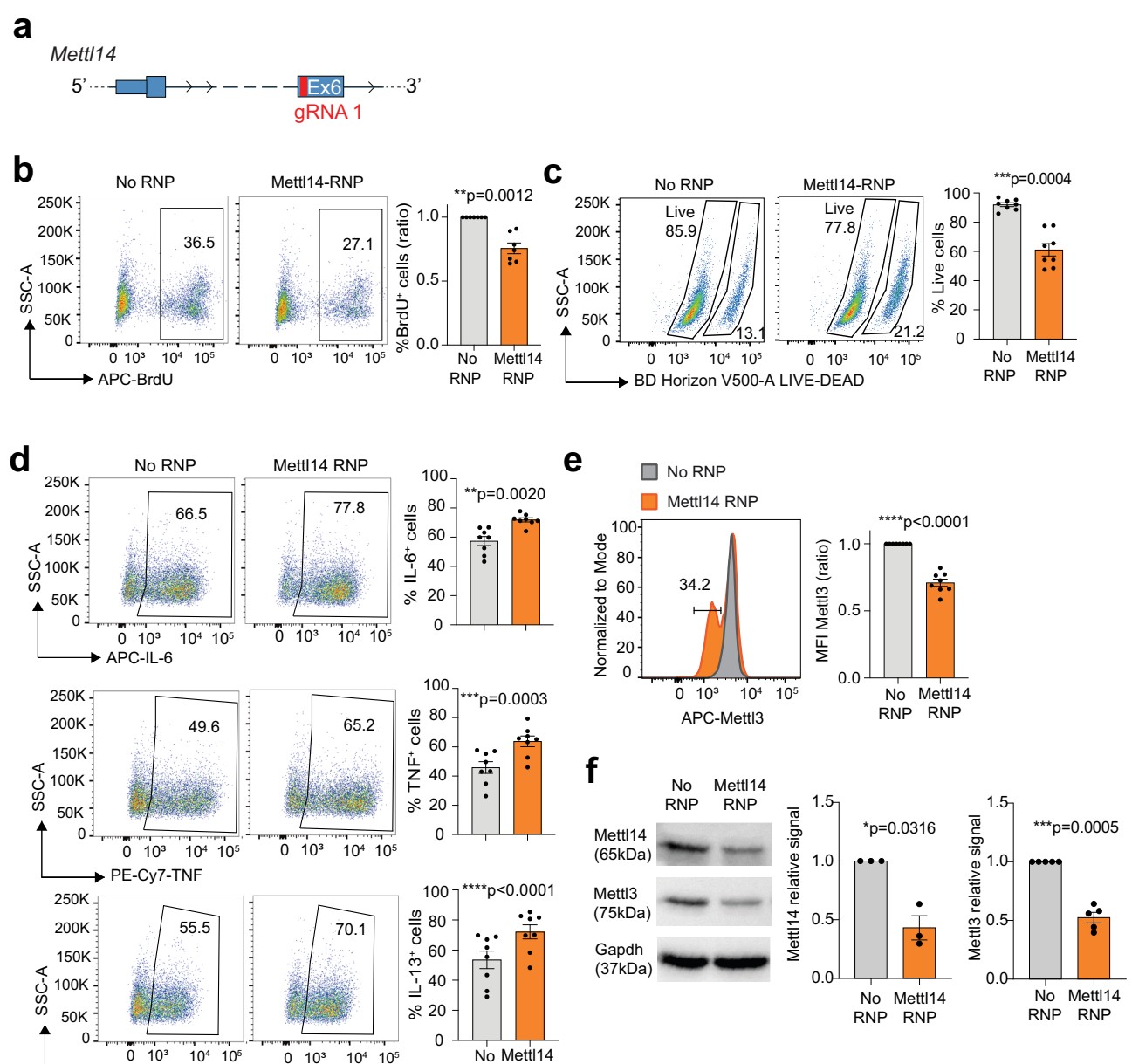

**Fig. 4 | Deletion of Mettl14 phenocopies Mettl3 deficiency. a** Schematic representation of the *Mettl14* locus with location of the gRNA. **b** Cell proliferation (measured by BrdU incorporation) in mast cells transfected with Mettl14 RNPs. *N* = 7 independent experiments. Mean ± SEM. Paired *t* test, two-tailed. **c** Cell viability in mast cells transfected with Mettl14 RNPs. Each dot represents one experiment. *N* = 8 independent experiments. Mean ± SEM. Paired *t* test, two-tailed. **d** Mast cells transfected with Mettl14 RNPs were stimulated with IgE and antigen complexes and cytokine expression was measured by intracellular staining. *N* = 8 independent experiments. Mean ± SEM. Paired *t* test, two-tailed. **e** Mettl14 depletion induces the loss of Mettl3 expression. Intracellular staining for Mettl3 was performed on mast cells transfected with Mettl14 RNPs. *N* = 8 independent experiments. Mean ± SEM. Paired *t* test, two-tailed. **f** Western blot for Mettl14, Mettl3 and Gapdh as control was performed on mast cells transfected with Mettl14 RNPs. *N* = 3-5 independent experiments. Mean ± SEM. Paired *t* test, two-tailed. Source data are provided as a Source data File.

Overall, m6A methylation by Mettl3 is associated to changes in the expression of genes involved in basic cellular processes.

**Effective gene editing by homology-directed repair in mast cells**
Our RNA-seq analysis was performed on cells transfected with RNPs at high efficiency, but still containing a remaining population of untargeted cells. To better investigate the mechanistic impact of Mettl3 on the mast cell transcriptome, we developed a system to allow us to select the cells that successfully underwent *Mettl3* gene deletion within a live mixed population. We therefore resorted to a knock-out/ knock-in approach based on mechanisms of homology-directed repair (HDR).

We designed a reporter construct to be used as donor DNA template, containing a ZsGreen reporter gene followed by a polyA-STOP cassette and flanked by homology arms of ~400 bp in length, targeting either the *Kit* (exon 2) or *Mettl3* (exon 1) gene (Supplementary Fig. 5a). The donor plasmid was either directly co-transfected together with RNPs, or it was used as a template to generate single-stranded DNA (ssDNA), that was then transfected with Cas9 RNPs. When targeting the *Kit* gene, we found that ssDNA provided more efficient and reproducible gene deletion, while at the same time correctly inserting the reported ZsGreen cassette in ~11% of the cells (Supplementary Fig. 5b). To confirm insertion specifically within the *Kit* locus, we sorted ZsGreen+

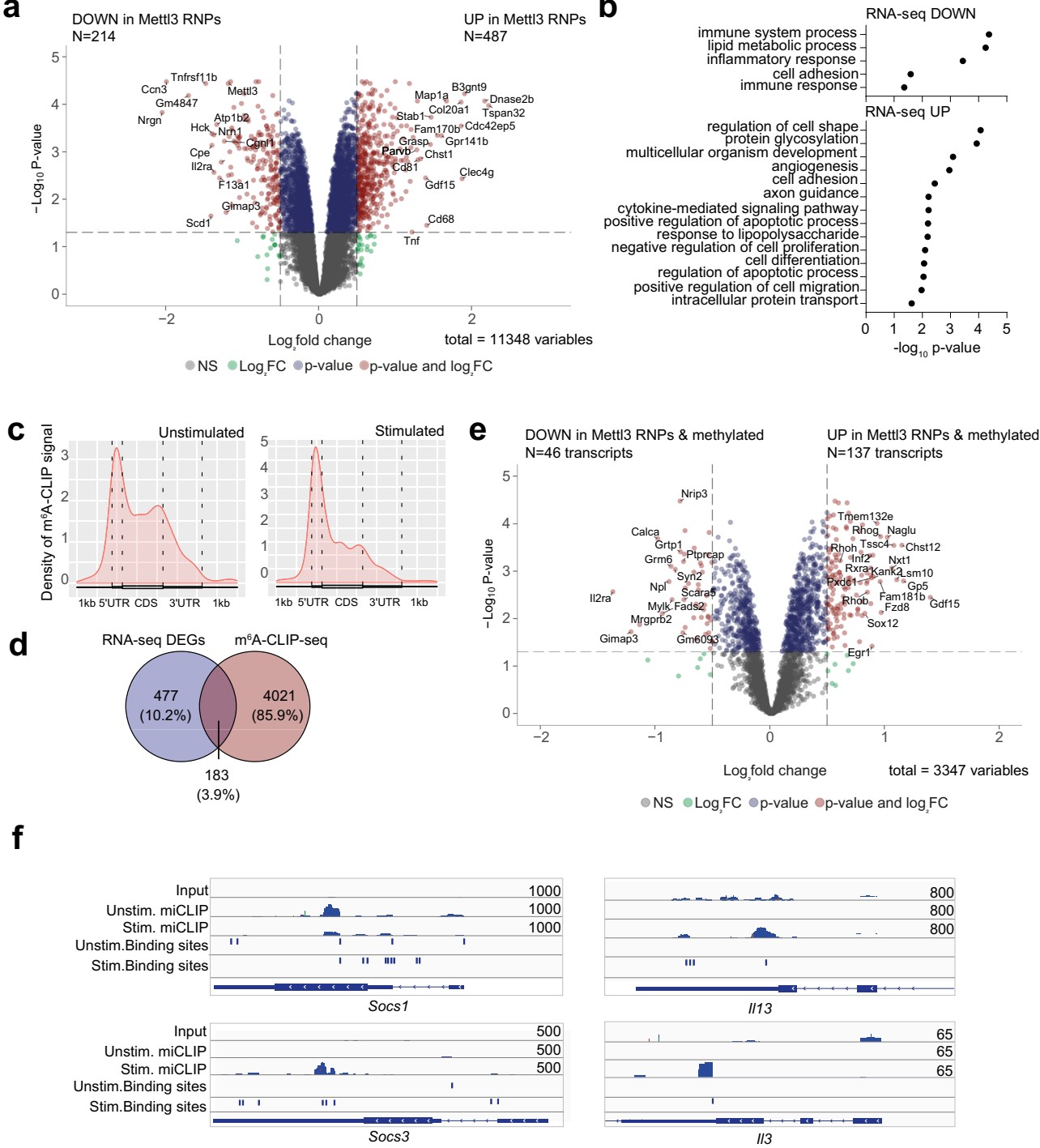

**Fig. 5 | Analysis of transcript methylation and abundance in mast cells. a** RNA-seq of wild-type and Mettl3-KO BMMCs. $N = 4$ independent biological replicates. RNAseq data were analyzed with DESeq2 R package. Wald test $P$ values were then adjusted for multiple testing using Benjamini and Hochberg correction. **b** GO terms associated to down- and up-regulated genes that were differentially expressed. Only categories containing at least 10 genes are shown. **c** miCLIP-seq of mast cells resting or stimulated with PMA and ionomycin. $N = 3$ independent biological replicates. The density of miCLIP signal is shown. **d** Venn diagram showing the overlap between differentially expressed genes (DEGs) identified by RNA-seq in (**a**) and methylated transcripts. **e** Volcano plot of DEGs identified in (**a**) that are also methylated. **f** Integrative Genomics Viewer tracks displaying miCLIP read distribution on representative transcripts. The read coverage is shown for merged biological replicates. The binding sites are identified by PureCLIP analysis incorporated with size-matched input control. Data were prepared from three biological replicates. Source data are provided as a Source data File.

c-Kit⁻, ZsGreen⁻ c-Kit⁻ and ZsGreen⁻ c-Kit⁺ cells and we extracted the genomic DNA. No ZsGreen⁺ c-Kit⁺ cells could be sorted in sufficient numbers, suggesting that very few cells, if any, inserted the ZsGreen reporter cassette on one allele while at the same time retaining one wild-type allele, and that the most frequent event was the deletion of

the *Kit* gene on both alleles. Analysis of the genomic DNA using PCR primers located across the expected region of recombination confirmed the presence of the ZsGreen-polyA-STOP cassette within the *Kit* locus only in ZsGreen⁺ c-Kit⁻ cells (Supplementary Fig. 5c). Next, we repeated the same knock-out/ knock-in procedure to delete the *Mettl3*

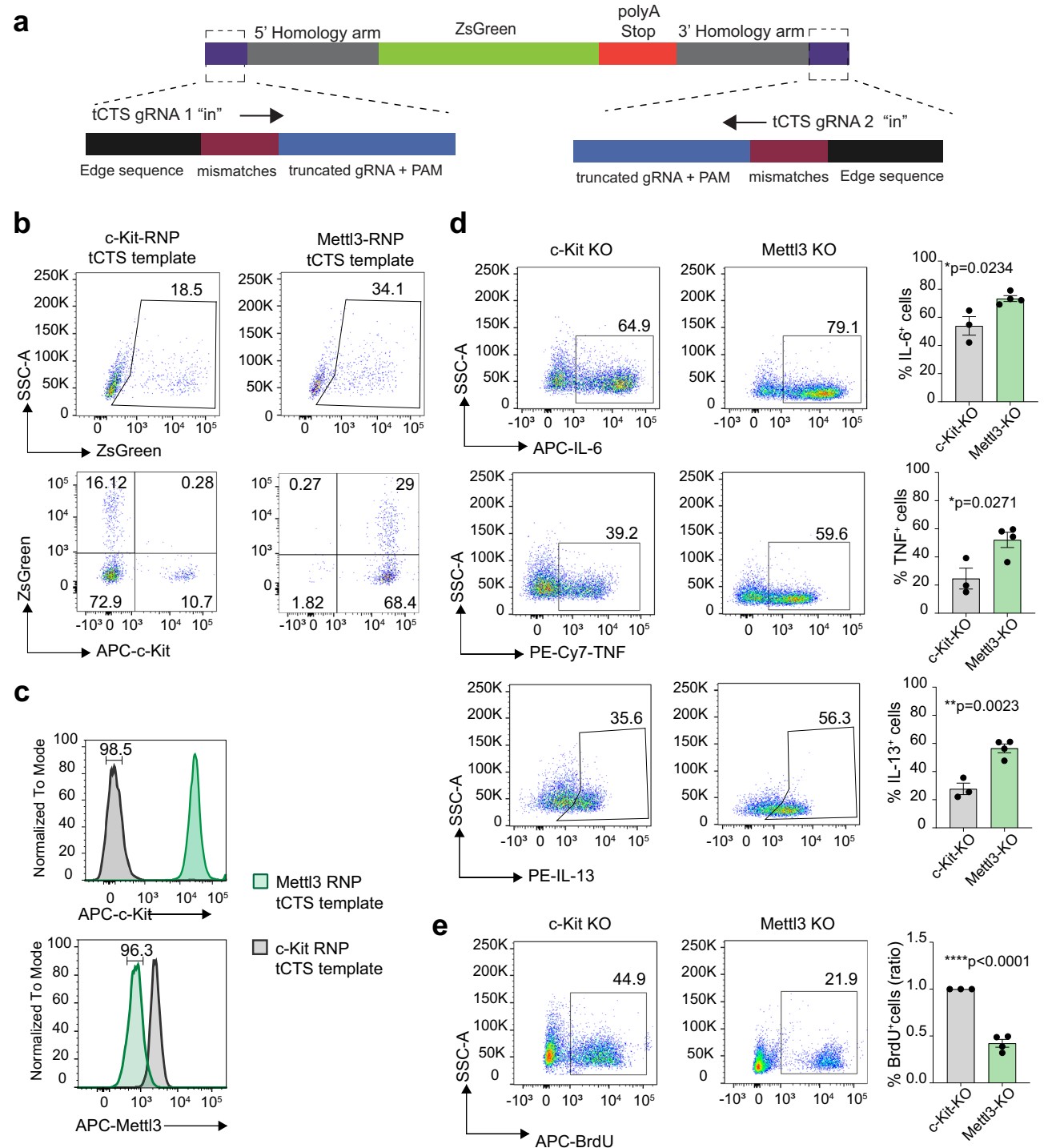

**Fig. 6 | Efficient gene replacement with truncated Cas9 target sequences templates in mast cells. a** Schematic representation of the tCTS HDR template used for gene replacement. **b** Mast cells were transfected with *Kit*- or *Mettl3*-targeting RNPs together with the tCTS template as in a). Expression of c-Kit and ZsGreen was measured 72 h after transfection. Representative of *N* = 6 experiments. **c** Cells transfected as in b) were sorted for ZsGreen and the expression of Mettl3 protein measured by intracellular staining. Representative of *N* = 3 experiments. **d** Cells as in (**c**) were stimulated with IgE-antigen complexes and cytokine expression was measured by intracellular staining. Left, one representative experiment. Right, *N* = 3-4 independent experiments. Mean ± SEM. Unpaired *t* test, two-tailed. **e** BrdU incorporation in cells as in (**c**). Left, one representative experiment. Right, *N* = 3-4 independent experiments. Mean ± SEM. Unpaired *t* test, two-tailed. Source data are provided as a Source data File.

gene while at the same time inserting the reporter cassette. We found that after transfection ~5% of the cells expressed ZsGreen (Supplementary Fig. 5d), and that the ZsGreen+ cells had increased cytokine expression by intracellular staining, a phenotype consistent with the one initially observed in *Mettl3*-depleted cells (Supplementary Fig. 5e).

However, the low percentage of correctly targeted cells limited the possibility of using these cells for further downstream analyses.

To improve the efficiency of genome editing, we added truncated Cas9 target sequences (tCTSs) to the ends of the HDR template (Fig. 6a). This was reported to enable shuttling of the template to the

cell nucleus, enhancing HDR efficiency[29]. Cas9 RNPs were further sta-bilized by the addition of polyglutamic acid[29]. Upon transfection of mast cells, the combination of these improvements led to a 6.8-fold increase in *Mettl3* gene targeting efficiency (from ~5% to ~34% ZsGreen+ cells) (Fig. 6b). To confirm specific deletion of the correct gene, we sorted the ZsGreen+ population from cells transfected with the *Kit*-targeting template (c-Kit KO cells) and *Mettl3*-targeting template (Mettl3 KO cells). Staining for Mettl3 confirmed ablation of the Mettl3 protein in Mettl3 KO cells, compared to c-Kit KO cells, and vice versa (Fig. 6c). Consistent with our previous results, Mettl3 KO cells showed significant increased expression of inflammatory cytokines and reduced proliferation (Fig. 6d, e), establishing a role for Mettl3 in modulating basal cell growth and in restraining mast cell responses upon stimulation with IgE and antigen.

### Mettl3 deficiency affects the expression of inflammatory transcripts

Expression of inflammatory cytokines is central to the outcome of immune responses. Since we found that cytokine transcripts were m⁶A-methylated in stimulated mast cells (Fig. 5f and Supplementary Data 2), to further investigate the mechanism(s) underlying the phe-notype observed in the absence of Mettl3 on inflammatory responses, we sorted c-Kit KO and Mettl3 KO cells generated using tCTS-HDR constructs, and we performed gene expression analysis using the NanoString "myeloid innate immunity" panel in cells either stimulated with IgE and antigen for 4 h or left resting (Supplementary Data 3), therefore complementing our RNA-seq dataset obtained from unsti-mulated cells. IgE and antigen complexes induced the expression of many inflammatory cytokines and chemokines in both c-Kit KO and Mettl3 KO cells, as expected (Fig. 7a, b). Direct comparison of c-Kit KO and Mettl3 KO mast cells revealed that Mettl3 targeting did not sub-stantially affect mast cell identity or activation state (Fig. 7c), as shown also by the largely unaltered expression of immune-related transcripts by RNA-seq in the absence of stimulation. Concordant with the fact that the most established functional outcome of m⁶A is the destabili-zation of the methylated transcripts[9], we observed that most differ-entially expressed inflammatory transcripts were upregulated in stimulated mast cells in the absence of Mettl3, and only very few were downregulated (Fig. 7d). Among the most differentially expressed genes were many transcripts encoding for cytokines and chemokines, including *Tnf*, *Il13* and *Il6*, a result consistent with the increased expression of the corresponding cytokines observed by intracellular staining in cells lacking Mettl3. Other upregulated cytokines included *Csf2*, *Il3* and *Il2* (Fig. 7e), although the latter was expressed at only very low levels in unstimulated cells. *Serpine1* encodes for a protein known to induce mast cell-dependent airway inflammation and tissue remo-deling in a mouse model of asthma[30], further highlighting a role for Mettl3 in restraining mast cell inflammation. Very few genes were downregulated in Mettl3 KO cells, and only modestly (Fig. 7f). The differential expression of some of the upregulated genes was validated in independent samples (Fig. 7g). Importantly, of the 45 most upre-gulated genes in stimulated Mettl3 KO cells, 26 (57%) were found to be methylated, as determined by miCLIP-seq (red asterisks in Fig. 7e). These data suggest an important role for m⁶A methylation in down-modulating the expression of inflammatory transcripts in mast cells.

### Mettl3 directly targets inflammatory transcripts and modulates their stability

Next, we asked whether the effect of Mettl3 on cytokine expression was direct. We cloned the 3'UTR of *Il13*, *Il6* and *Tnf* downstream a luciferase reporter gene. Co-transfection of the reporter vectors together with a Mettl3 expression plasmid significantly reduced luci-ferase expression from all constructs (Fig. 8a), pointing towards a direct role for Mettl3 in modulating transcript stability and/or trans-lation. To assess whether the methyltransferase enzymatic activity of

Mettl3 was indeed necessary to impact cytokine expression, we mutated the DPPW motif (amino acids 395-398) to APPA, which is known to impair Mettl3 methyltransferase activity[31,32]. Both proteins were highly expressed in transduced mast cells (Fig. 8b). Over-expression of wild-type Mettl3 in mast cells by lentiviral transduction led to the expected reduction in cytokine expression (Fig. 8c, d and Supplementary Fig. 6a). The overexpression of the catalytically inac-tive APPA mutant showed instead variable effects, likely due to the expression of high basal levels of endogenous wild-type Mettl3, that might be competing with the overexpressed protein for the methyl-transferase complex. However, for the most part we observed no sig-nificant reduction in cytokine production, suggesting that the catalytic activity of Mettl3 is required to modulate the expression of inflam-matory transcripts (Fig. 8c, d). Since the presence of m⁶A in mature, cytoplasmic mRNAs is predominantly associated with reduced stability of the methylated transcripts, we measured the stability of the endo-genous *Il13* and *Tnf* transcripts after treatment of wild-type or Mettl3-KO cells with actinomycin D (Fig. 8e). We found that in absence of Mettl3 the stability of both endogenous transcripts was increased, establishing Mettl3 as a key regulator of the expression of inflamma-tory cytokine in mast cells.

IL-13 expression and *Il13* transcript stability were strongly affected by Mettl3 deletion or overexpression. To determine if the regulation of IL-13 expression depended on the enzymatic activity of Mettl3 on the 3'UTR of the *Il13* transcript we performed luciferase reporter assays using wild-type Mettl3 or its APPA mutant. Since the variability in cytokine expression observed in mast cells overexpressing the APPA mutant was likely to be due to the presence of high levels of endo-genous wild-type Mettl3, we first deleted the endogenous *METTL3* gene in HEK293T cells by CRISPR-Cas9 (Fig. 8f), and then used these cells for luciferase assay. In these conditions of reduced endogenous METTL3 expression, luciferase assays using a reporter construct con-taining the *Il13* 3'UTR showed reduced expression upon co-transfection of wild-type Mettl3, but not its APPA mutant (Fig. 8f), further suggesting a direct role for the catalytic activity of Mettl3 in modulating *Il13* expression/ stability through its 3'UTR.

Most mammalian mRNAs have four or less sites that are actually methylated, and only in a proportion of mRNA molecules[33]. By miCLIP-seq, we found that the *Il13* transcript contained regions within its 3'UTR that were methylated in murine mast cells. Within these methylated regions, we identified and mutated three putative DRACH motifs (D = G/A/U, R = G/A, H = A/U/C), since they are the potential sites of m⁶A methylation in the *Il13* transcript (Fig. 8g). We then used a tethered reporter construct, containing the 3'UTR of *Il13* followed by 5 box B sites. The box B DNA results in a stem-loop structure in the nascent RNA that is recognized with high affinity by the λ peptide, a 22 amino acids sequence forming the RNA-binding domain of the λ phage antiterminator protein N[34,35]. Such interaction has already been extensively used to tether desired protein activities to the 3'UTR of genes[36,37]. This construct was co-transfected in HEK293T lacking endogenous METTL3 expression, together with a plasmid expressing Mettl3 fused to the λN peptide. We found that the expression of λN Mettl3 was sufficient to reduce the expression of a reporter containing the wild-type *Il13* 3'UTR. Mutation of the DRACH motifs proximal to the stop codon in any combination was sufficient to ablate the negative effect of Mettl3 transfection on the *Il13* 3'UTR, confirming that the effect of Mettl3 is direct and mediated through the DRACH sites (Fig. 8h). Overall, our findings indicate that Mettl3 restrains IL-13 expression in mast cells by acting directly on the *Il13* 3'UTR in an enzymatic activity-dependent manner.

## Discussion

The role of the mRNA methyltransferase complex in mast cells was so far unexplored. One of most prominent phenotypes that we observed in mast cells lacking Mettl3 was their increased responses to acute

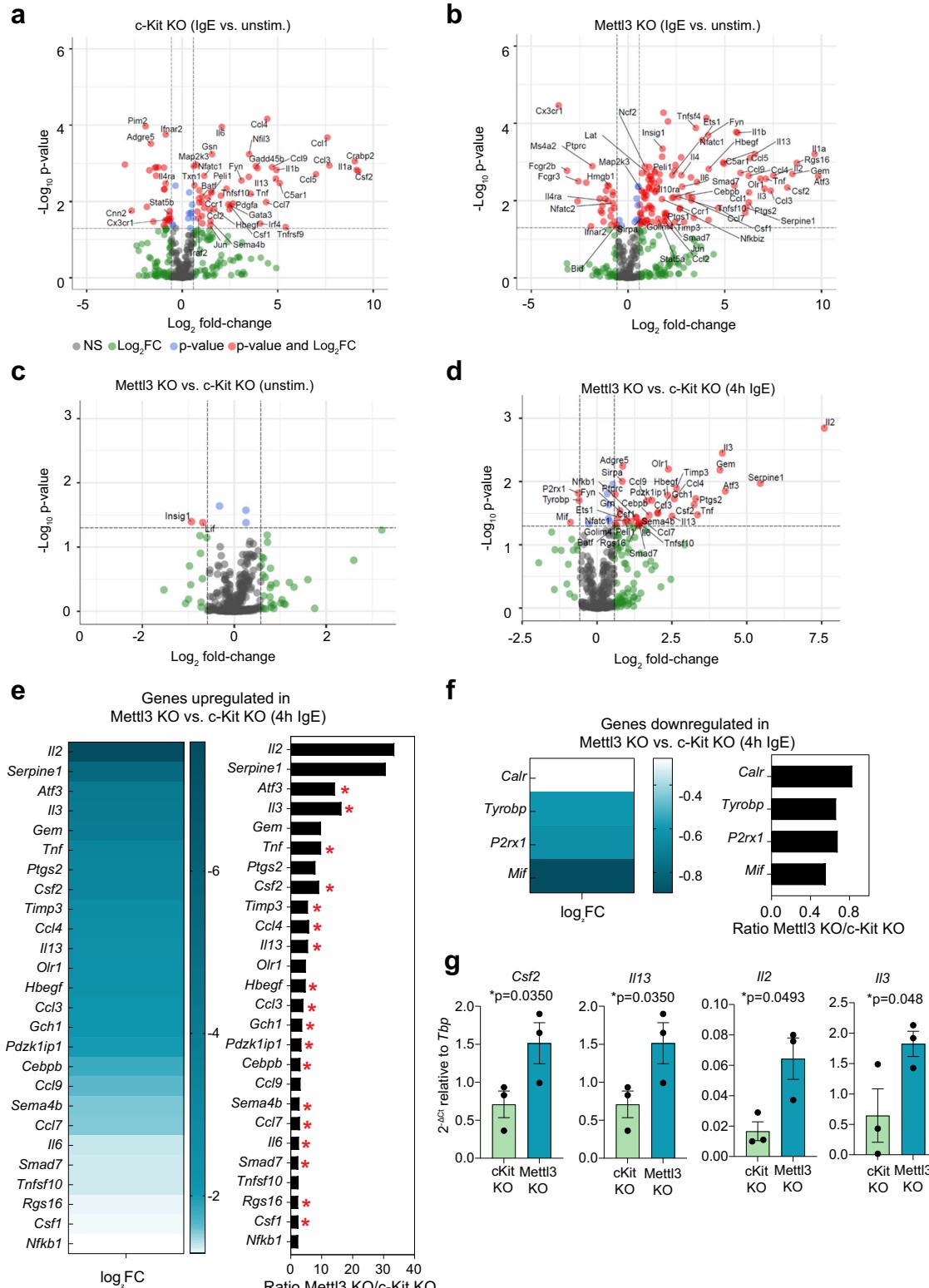

**Fig. 7 | Mettl3 modulates the expression of inflammatory transcripts in mast cells. a–d** Cells lacking either c-Kit or Mettl3 expression were stimulated with IgE+antigen complexes and the expression of inflammatory transcripts was measured using the Nanostring Myeloid Innate Immunity Panel. Shown are individual scatter plots comparing unstimulated and stimulated c-Kit KO and Mettl3 KO cells. $N = 3$ independent biological replicates. Two-tailed $t$ test, without further $p$ value adjustment. **e** Genes that are most differentially expressed (upregulated in absence of Mettl3) between stimulated c-Kit KO and Mettl3 KO cells. Both the $\log_2$ fold-change (FC) and the ratio are shown. The red asterisks indicate transcripts that were found to be methylated by miCLIP-seq in stimulated mast cells. **f** Genes that are most differentially expressed (downregulated in absence of Mettl3) between stimulated c-Kit-deleted and Mettl3-deleted cells. Both the $\log_2$FC and the ratio are shown. **g** Validation of gene expression results using an independent set of experiments. $N = 3$, each dot represents one experiment. Mean ± SEM. Paired $t$ test, two-tailed. Source data are provided as a Source data File.

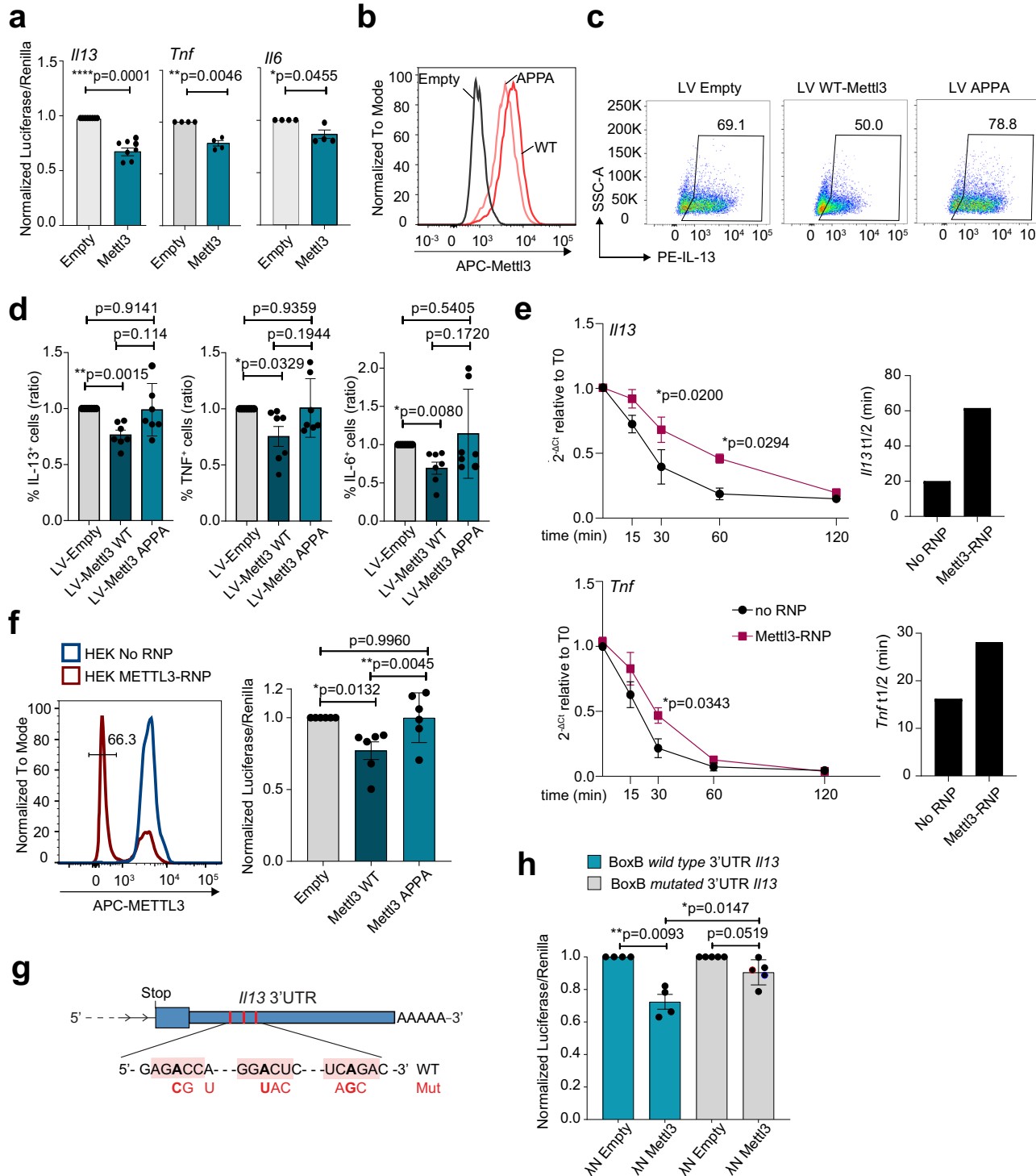

**Fig. 8 | Mettl3 deletion stabilizes inflammatory transcripts in mast cells.**
**a** HEK293T cells were transfected with luciferase reporter plasmids containing the indicated 3'UTRs, with or without Mettl3. *N* = 4-8, each dot represents one experiment. Mean ± SD. Paired *t* test, two-tailed. **b** Mast cells were transduced with lentiviruses to overexpress either wild-type (WT) or APPA Mettl3. After selection and expansion, protein overexpression was detected by intracellular staining for Mettl3. One representative experiment of *N* = 4. **c, d** Cells expressing either wild-type or APPA Mettl3 were stimulated with IgE and antigen complexes and cytokine expression was measured by intracellular staining. *N* = 7 independent experiments. Mean ± SD. Paired *t* test, two-tailed. **e** Mast cells were transfected with Cas9 RNPs to delete Mettl3, after which they were stimulated with IgE and antigen complexes and treated with actinomycin D for the indicated times. Expression of *Il13* and *Tnf* was measured by RT-qPCR and the

half-life of the transcripts (t1/2) was calculated. Mean ± SEM. Two-way ANOVA. **f** METTL3-KO HEK293T cells (intracellular staining to show METTL3 deletion in these cells is shown on the left) were transfected with a luciferase reporter plasmid containing the *Il13* 3'UTR, with or without wild-type or APPA Mettl3. *N* = 6, each dot represents one experiment. Mean ± SD. Paired *t* test, two-tailed. **g** Schematic representation of the 3'UTR of the *Il13* mRNA with indicated the three putative DRACH sites that were mutated as shown in red. The "A" nucleotides in bold correspond to the putative methylation site within the DRACH consensus (shaded). Stop: stop codon. **h** METTL3-KO HEK293T cells were transfected with a luciferase reporter plasmid containing the *Il13* 3'UTR with BoxB sites, wild-type or mutated in the DRACH sites, together with λN Mettl3. *N* = 4-5, each dot represents one experiment. Mean ± SD. Paired or unpaired *t* test, two-tailed. Source data are provided as a Source data File.

stimulation. This was at least in part linked to the direct effect of Mettl3 on the half-life of inflammatory transcripts like *Tnf* and *Il13*. Both TNF and IL-13 are central cytokines that mediate mast cell effector functions and tissue inflammation, as shown in mice lacking either *Tnf* or *Il13* specifically in mast cells[2,38]. Indeed, mast cell-derived TNF was shown to be crucial for neutrophil extravasation and recruitment in various models of skin inflammation[2]. Similarly, mast cell-derived IL-13 in response to mechanical skin injury (scratching) influenced the phenotype of dendritic cells in the skin, inhibiting Th1-type cell responses in models of dermatitis[38]. Understanding how these cytokines are regulated can therefore lead to a better comprehension of mechanisms underlying inflammatory diseases.

By focusing on the 3'UTR of the *Il13* transcript we found that mechanistically the effect of Mettl3 on this transcript was mediated by specific DRACH sites in the 3'UTR and depended on the enzymatic activity of Mettl3. In general, cytokine transcripts can be regulated by multiple RNA-binding proteins (RBPs) in a complex interaction landscape able to provide temporal control to cytokine expression. However, the RBPs involved in the regulation of cytokine mRNA stability are only partially known, and the extent of functional cooperation or interference among the different RBPs in this process is also not fully understood. For instance, expression of TNF is prominently regulated at the post-transcriptional levels by the RBP tristetraprolin (TTP), and whether altered methylation of *Tnf* also affects binding of other RBPs remains to be established. Indeed, the presence of m6A reduces the ability of the methylated RNA to form secondary structures, thereby potentially affecting binding of other RBPs, even those that do not recognize m6A directly. We found that the m6A reader Ythdf2 could be detected as bound to the *Il13* 3'UTR in stimulated cells (Supplementary Fig. 6b–d). However, other RBPs may preferentially bind m6A-methylated RNA because of its increased accessibility, rather than m6A recognition, making difficult to discern between these different possibilities when studying mRNA:RBPs interactions[8,39,40]. Similar to other systems[20,28], we found that the transcripts encoding the negative regulators of cytokine signaling *Socs1*, *Socs3* and *Cish* were methylated in mast cells. However, we did not observe the anticipated stabilization of these transcripts upon deletion of Mettl3, suggesting that the m6A-mediated mechanisms of regulation of these transcripts are not predominant in mast cells, at least in the absence of acute stimulation.

Consistent with the deletion of Mettl3, ablation of Wtap and Mettl14 also led to reduced mast cell proliferation. This is consistent with observations from other cell types, like T lymphocytes. In these cells, both Wtap and Mettl3 deficiency impaired Treg functionality, causing colitis[20]. However, the reduction of surface FcεRI expression appeared to be Wtap-specific. Wtap is not only an essential component of the m6A RNA methyltransferase complex, but it is also involved in regulating alternative splicing of specific targets (such as genes with short introns and high GC content[21]), possibly in part through its interaction with the Wilm's tumor 1 (WT1) protein[22]. However, a final answer on the extent of the putative m6A-independent functions of Wtap is currently out of reach, since it would require uncoupling the m6A-related and -unrelated functions of Wtap. It is also worth to point out that we did not measure global m6A changes in each individual experimental setup, therefore it remains possible that levels of m6A may be depleted at variable extent depending on the experimental system (RNAi, CRISPR-Cas9 with or without HDR, deletion of Mettl3 vs. Mettl14).

Deletion of Mettl14 phenocopied the effect of deleting Mettl3 in mast cells also in terms of cell survival and cytokine production. However, it also affected the expression of Mettl3 itself, leading to its depletion. The expression of other components of the methyltransferase complex was also affected, suggesting that changes in the stoichiometry of the various components affect the complex in its entirety. Although we were able to show that at least on some transcripts (most notably *Il13*) the effect of Mettl3 is direct and mediated

by m6A methylation at specific DRACH sites, the effect of deleting especially other components of the complex may be linked, at least in part, to the indirect reduction in Mettl3 expression.

Overall, with this work, we were able to establish the importance of the m6A methyltransferase complex in modulating mast cell functions, at least in part mediated by the regulation of cytokine transcript stability. The future challenge will be to perform all the meticulous work required to understand how and why each transcript is bound and regulated by its own 'constellation' of RBPs and how these proteins interact with each other to achieve such combinatorial post-transcriptional regulation[41].

## Methods

### Mice, murine mast cells and human mast cell lines

Bone marrow-derived mast cells (BMMCs) were differentiated from the bone marrow of 6-8 week old C57BL/6 mice by culturing whole bone marrow cells for at least 4 weeks (until the percentage of FcεRIα+ c-Kit+ cells was ~95%) in the presence of IL-3 (made in-house), as previously described[3,4]. Kit^Wsh/W-sh mice lacking mast cells were obtained from The Jackson Laboratory[19]. When indicated, hematopoietic precursors were enriched from the bone marrow by negative magnetic selection using a lineage-cell depletion kit (Miltenyi Biotec). To generate mast cells lacking Wtap, bone marrow from *Wtap*fl/fl mice[20] were transduced with a lentiviral vector expressing the Cre recombinase fused to an EGFP reporter. EGFP+ cells were then sorted for downstream analyses. Peritoneal-derived mast cells (PMCs) were obtained from mouse peritoneal cells isolated by intraperitoneal lavage and enriched for c-Kit expression using magnetic beads (Miltenyi Biotec). Cells were expanded for at least 1 week in the presence of IL-3 and 30 ng/ml recombinant SCF (Peprotech). All animal studies were performed in accordance with Swiss Federal Veterinary Office guidelines and approved by the Cantonal animal experimentation committee, Dipartimento della Sanità e della Socialità Cantone Ticino (authorization number TI10/19). Housing conditions for the animals were 12 h dark/light cycles, 20-24 °C temperature and 50-65% humidity. The HMC-1.1 and 1.2 human mast cell lines were kindly provided by Joseph Butterfield and were cultured as described[42].

### Mast cell activation

BMMCs and PMCs were stimulated for 4 h with the following: 1 μg/ml IgE-anti DNP antibody (Sigma) and 0.2 μg/ml HSA-DNP (Sigma) or 20 nM PMA (Sigma) and 2 μM ionomycin (Sigma), and/or 0.1 ng/ml recombinant IL-33 (Biolegend). When used in combination, IL-33 and HSA-DNP were added simultaneously. HMC-1.1 and 1.2 cells were stimulated for 4 h with 20 nM PMA and 2 μM ionomycin.

### Flow-cytometry analysis of surface markers and intracellular cytokine staining

Surface staining was performed using the following antibodies: CD117 (c-Kit)-APC or APC/Cy7, FcεRIα-PE, CD11b (Mac-1)-Pacific Blue (all from Biolegend) and Ly-6G (Gr-1)-PE-Cyanine7 (eBioscience). All antibodies for surface staining were used at a dilution of 1:200. For intracellular cytokine staining, cells were stimulated with 1 μg/ml IgE−anti-DNP antibody and 0.2 μg/mL HSA-DNP antigen (both from Sigma) for 3.5 h, with the addition of brefeldin A in the last two hours of stimulation, as previously described[3,4]. The following antibodies were used: IL-6-PE or APC, TNF-α-PE/Cy7 (both from Biolegend) and IL-13-PE (eBioscience). All antibodies used in this study are listed in Supplementary Table 1. Flow-cytometry data were collected on a FACS Symphony A5 or Fortessa (BD Biosciences) and analyzed by FlowJo version 10 (BD Biosciences).

### Intracellular staining for components of the m6A complex

Intracellular staining for Mettl3, Wtap and Virma was performed as previously described[20]. Briefly, cells were washed once with PBS and

stained with LIVE/DEAD-Fixable Aqua Dead Cell Stain (ThermoFisher Scientific) for 20 min at room temperature. To fix and permeabilize cells, the Foxp3/Transcription Factor Staining buffer (eBioscience) was used according to manufacturer's instructions. Following permeabilization, cells were incubated 40 min at room temperature with either 4 μg/ml of anti-METTL3 antibody (clone EPR18810, Abcam) or 10 μg/ml of anti-Wtap monoclonal antibody (clone 4A10G9, Proteintech). The anti-Virma antibody (clone D4N8B, Cell Signaling Technology), was used at 0.7 μg/ml. Secondary antibody staining was carried out for 30 min at room temperature using either Alexa Fluor 647 anti-rabbit IgG (H + L) (Sigma) (for Mettl3 and Virma) or Alexa Fluor 647 (or 594) goat anti-mouse IgG (H + L) (ThermoFisher Scientific) (for Wtap staining). All secondary antibodies were used at a 1:400 dilution.

## Degranulation assay

Degranulation assay was performed as previously described[3]. Briefly, 0.5-1.5 ×10^6 BMMCs were pre-treated with 1 μg/ml IgE-anti DNP antibody overnight at 37 °C. Cells were then resuspended in Tyrode's buffer (10 mM HEPES, 129 mM NaCl, 5 mM KCl, 15.05 mM BSA, 1.8 mM CaCl_2, 2 mM MgCl_2, 5 mM glucose) and stimulated in a 96-well U-bottom plate with 0.2 μg/ml HSA-DNP for 1 h at 37 °C. 20 nM PMA and 2 μM ionomycin were used in parallel as positive control of the stimulation. 10 μl of supernatant were collected in a flat-bottom 96-well plate while cell pellets were lysed in 0.5% Triton X-100 in Tyrode's buffer. 10 μl cell lysate were collected in a second flat-bottom 96-well plate. 50 μL β-hexosaminidase substrate (4-nitrophenyl N-acetyl-β-D-glucosaminide 4 mM, Sigma) were then added to flat-bottom 96-well plates for 1 h at 37 °C and the reaction was stopped with 150 μL of 0.2 M glycine (pH 10.7). Finally, absorbance was read at 405 nm and the percentage of degranulation was calculated as the ratio between the absorbance of the supernatants on the total absorbance of the supernatants plus cell lysates.

## Reverse-transcriptase quantitative PCR (RT-qPCR)

Total RNA was isolated using TRIreagent (MRC) followed by purification using the Direct-zol RNA mini prep kit (Zymo Research). qScript cDNA SuperMix (Quanta Bioscience) was used for retrotranscription and PerfeCTa SYBR Green FastMix (Quanta Bioscience) was used to amplify target genes. Primer sequences are listed in Supplementary Table 2. qPCR reactions were run on an ABI 7900HT Fast Real-Time PCR system (Applied Biosystems) or a QuantStudio 3 Real-Time PCR System (ThermoFisher Scientific). Data analysis was performed using the $2^{-\Delta\Delta Ct}$ or $2^{-\Delta Ct}$ method.

## Cell proliferation and viability

BrdU incorporation assay was used to measure mast cell proliferation using an APC BrdU Flow Kit (BD Bioscience), according to the manufacturer's instructions and as previously described[3,4]. Cells were left to incorporate BrdU for 8–16 h at 37 °C. For cell viability assay, cells were washed once with PBS and stained with LIVE/DEAD-Fixable Aqua Dead Cell Stain (ThermoFisher Scientific) for 20 min at room temperature.

## Western blots

BMMCs were lysed in RIPA buffer (10 mM Tris-HCl pH 8.0, 1 mM EDTA, 1% Triton X-100, 0.1% sodium deoxycholate, 0.1% SDS, 140 mM NaCl) supplemented with a cocktail of protease inhibitors (Sigma). Protein concentration was measured using a Pierce BCA assay (ThermoFisher Scientific). Protein samples (50 μg) were run on 8% SDS/polyacrylamide gels and blotted on a PVDF membrane using a wet transfer system and a methanol-based transfer buffer (20 mM Tris, 150 mM glycine, 20% methanol). Blocking was performed with 5% milk in TBST (5 mM Tris pH 7.3, 150 mM NaCl, 0.1% Tween-20) for 60 min at room temperature with gentle shaking. Specific for Mettl14 detection, blocking was performed with 5% BSA in TBST. Immunodetection was performed using primary antibodies at a 1:1000 dilution overnight at 4 °C. The specific antibodies used are listed in Supplementary Table 1. This was followed by washing and incubation with the corresponding HRP-conjugated secondary antibody (ThermoFisher Scientific) at a 1:10,000 dilution 1 h at room temperature. Blot development was performed using the ECL Prime Western Blotting Detection Reagent (Amersham) and analyzed with a blot imager (GE, Amersham Imager 680). Protein band intensity was quantified using the ImageJ software, version 1.53e.

## Plasmids

Plasmids were generated using standard cloning techniques. The human *METTL3* gene was subcloned from Addgene vector 53739[23] into pScalps (Addgene vector 99636)[4], while the murine *Mettl3* gene was amplified from cDNA obtained from BMMCs, cloned into the pScalps vector and finally tagged with FLAG-HA. To generate the Mettl3-APPA mutant, residues D395 and W399 of Mettl3 were mutated to alanines using the Quick Change II XL site-directed mutagenesis kit (Agilent) according manufacturer's instructions. The luciferase reporter plasmids containing the full-length 3'UTR of the cytokine genes *Il13*, *Il6* and *Tnf* were generated by PCR using genomic DNA from BMMCs as a template and cloned downstream of the luciferase reporter gene in the pmirGLO dual luciferase vector (Promega). For tethering assays, the lambda (λ) N peptide was cloned at the N-terminus of Mettl3, while 5 box B binding sites were cloned downstream the *Il13* 3'UTR. Five box B sequences were shown to be sufficient to induce a strong effect in reporter constructs[36]. Mutations of the DRACH motifs within the 3'UTRs of *Il13* were introduced using the Quick Change II XL site-directed mutagenesis kit (Agilent) according to manufacturer's instructions. The vector expressing the EGFP-Cre fusion protein was generated by subcloning EGFP-Cre from pLV-EGFP-Cre (Addgene number 86805)[43] into pScalps. To generate the templates for homology-directed repair (HDR) experiments, pBluescript II Ks(+) was used to clone the ZsGreen reporter gene from pLVX-EF1α-IRES-ZsGreen1 (Clontech) followed by the addition of the SV40 PolyA-Stop sequence from the pCAG-b3STOPb3-ZsGreen plasmid (Addgene number 51266)[44]. The resulting vector was then used to clone homologous arm sequences from the *Kit* and *Mettl3* loci, to generate the final donor templates for homologous recombination as previously described[29,45].

## Lentivirus preparation and mast cell transduction

HEK293T cells were transfected with a lentiviral vector together with the packaging vectors psPAX and pMD2.G (Addgene plasmids 12260 and 12259 by Didier Trono). Lentiviral particles were purified from supernatant of transfected HEK293T cells by sucrose gradient (10 mM Tris-HCl pH 7.5, 100 mM NaCl, 1 mM EDTA, 25% sucrose) followed by ultracentrifugation (2.5 h, 1,000,000 × g at 4 °C). Alternatively, viral particles were concentrated using a PEG-8000 solution as described[46]. 25×10^4 cells were seeded in a 48-well plate and 10-25 μl of virus per transduction were used. After at least 72 h, GFP-expressing cells were sorted using a FACSaria (BD Bioscience) or selected by puromycin (2 μg/ml for 48-72 h).

## siRNA transfection

siRNA transfection was performed using the 100 μl Neon Transfection System Kit (ThermoFisher Scientific). Briefly, 1×10^6 differentiated BMMCs (week 4-6 of in vitro culture) were washed once with PBS and resuspended in 100 μl of Buffer R. 100 μl of cell suspension were added to 200 pmol of Mettl3 custom siRNA or 200 pmol of siGLO fluorescent oligonucleotide transfection control (all from Dharmacon). SiRNA sequences are listed in Supplementary Table 2. Cells were electroporated with one pulse at 1600V and 30 ms of width. After transfection, cells were kept for 24 h in antibiotic-free cell culture medium prior to downstream analyses.

### Transfection of CRISPR-Cas9 ribonucleoproteins (RNPs)

One or two different crRNAs were selected per gene to be targeted (Supplementary Data 4). 400 pmol of each crRNA were combined with 400 pmol of tracrRNA-ATTO-550 in 10 µl reaction in Nuclease Free Duplex buffer (IDT). Oligonucleotides were subsequently annealed by boiling and cool-down to room temperature to generate gRNA complexes. Cas9 RNPs were prepared immediately before transfection by combining 0.75 µl of each gRNA with 1.5 µl TrueCut Cas9 Protein v2 (5 µg/µl, ThermoFisher Scientific) in a total volume of 3 µl and incubating 20 min at room temperature. Transfection was performed using the 10 µl Neon transfection System Kit (ThermoFisher Scientific). Briefly, $1 \times 10^6$ cells were washed once with PBS and resuspended in 10 µl of Neon electroporation Buffer R. Cells were added to RNPs and incubated 5 min at room temperature. Electroporation was performed using one pulse at 1600V and 30 ms of width for BMMCs, and one pulse at 1700V and 20 ms of width for HEK293T and HMC 1.1 and 1.2 cells.

### T7 endonuclease I assay

Genomic DNA was isolated from $5 \times 10^6$ to $10 \times 10^6$ mast cells using the DNeasy Blood & Tissue Kit (Qiagen). T7 endonuclease I assay to screen for mutations at selected targeted genes was carried out as previously described[47]. Briefly, PCR primers (Supplementary Table 2) were designed to amplify a DNA fragment of ~1000 bp around the area targeted by the gRNAs using the high fidelity KOD Hot Start DNA polymerase (Novagen). Then, 15 µl of the PCR reaction were denatured and reannealed. Finally, 100 U of T7 endonuclease I enzyme (New England Biolabs) were added to each reaction, incubated at 37 °C for 15 min and run on an agarose gel to assess the extent of digestion of the parental band.

### Preparation of double- and single-strand DNA (dsDNA and ssDNA) templates for homology-directed repair (HDR)

pBluescript plasmids containing a ZsGreen-PolyA-Stop cassette flanked by homologous arms were used to generate templates for HDR. The homology arms corresponded to regions upstream and downstream the gRNA target sites as previously described[45]. These plasmids were either directly electroporated as donors of dsDNA or used as templates to generate ssDNA. To generate ssDNA templates, PCR primers were designed to amplify ~400 bp of homology arms flanking the gRNA target sites. The Guite-it Long ssDNA Production system (TakaraBiotek) was then used according to manufacturer's instructions. For truncated Cas9 target sequences (tCTS) templates, truncated Cas9 target sequences were added at the ends of the HDR templates by PCR, as previously described[29]. PCR reactions were carried out using the PrimeSTAR Max DNA polymerase (TakaraBiotek) and NucleoSpin Gel and PCR Clean-UP were used to purify PCR amplicons. HDR template were concentrated to 1 µg/µl using a speed-vac concentrator. All HDR template sequences are listed in Supplementary Data 4.

### Gene replacement by CRISPR-Cas9

For gene replacement based on HDR templates, 5 µg of plasmid DNA templates or 1 µg of ssDNA were used. For tCTS PCR templates, 0.75 µl of each gRNA (2 gRNAs in total) were premixed with 150 µg of poly-L-glutamic acid (PGA, Sigma) as previously described[29]. 1.5 µl of TrueCut Cas9 Protein v2 (5 µg/µl, Thermo Fisher Scientific) were then added to the gRNA-PGA complexes and incubated 20 min at room temperature before addition of 750 ng of tCTS PCR template. The different RNP-HDR template complexes were then added to $1 \times 10^6$ cells in buffer R and electroporation was performed using the 10 µl Neon transfection System Kit (ThermoFisher Scientific) using one pulse at 1600V and 30 ms width. When needed, ZsGreen⁺ cells were then sorted using a FACSaria (BD Bioscience).

### Expression of inflammatory transcripts by digital profiling

Cells were either left resting or were stimulated for 4 h with IgE and antigen complexes. Purified RNA (40 ng) was hybridized to the NanoString nCounter Myeloid Innate Immunity Panel codeset for 16 h at 65 °C. Following hybridization, 30-35 µl of sample were added to the nCounter cartridge and analyzed using an nCounter SPRINT Profiler according to manufacturer's instructions. Data analysis was carried out using the nSolver Advanced Analysis Software v4.0 (NanoString Technologies). Differentially expressed genes were considered significant when $p$ value ≤ 0.05 and fold change ≥1.5.

### RNA-sequencing

Differentiated BMMCs were transfected with Mettl3 RNPs (four independent biological replicates). After verifying (by intracellular staining) that a minimum of 70% Mettl3 depletion was achieved, total RNA was extracted using Zymo-Spin IICR columns (Zymo Research). After poly-A mRNA enrichment, sequencing of Tecan Revelo mRNA libraries using Illumina Novaseq 6000 (2 ×50-bp reads) was outsourced to the Next Generation Sequencing Platform at the University of Bern (Switzerland). Quality control was performed using fastqc v.0.11.9 and RSeQC v.4.0.0[48]. Reads were mapped to the reference genome (Mus_musculus.GRCm39.107) using HiSat2 v.2.2.1[49]. Counts were generated using featureCounts v.2.0.1[50] and corrected for batch effects using the removeBatchEffect function in R[51]. Differential expression analysis was performed using the Omics Playground platform[52]. Gene ontology (GO) analysis was performed using DAVID[53]. Data visualization was performed with RStudio version 4.1 and Integrative Genomics Viewer (IGV) 2.16.0.

### Luciferase assays

HEK293T cells ($1 \times 10^6$) were seeded in a 6-well plate. After 24 h, cells were transfected with 3.5 µg of pmirGLO plasmid containing the 3′UTR of Il13, Il6 or Tnf and 1 µg of pScalps-FLAG-HA-Mettl3 or empty control plasmid. After 36-48 h, cells were lysed and analyzed using the Dual-Luciferase Reporter Assay System (Promega) according to the manufacturer's protocol. Luciferase activity was measured using a GloMax Luminometer (Promega). HEK293T cells lacking METTL3 were also used for luciferase assays when indicated. Briefly, cells were transfected with Cas9 RNPs to delete human METTL3 using gRNAs listed in Supplementary Data 4. Four days after transfection, and after measuring the extent of METTL3 deletion by intracellular staining (~60-70% deletion), $1 \times 10^6$ METTL3-KO HEK cells were seeded in a 6-well plate. For tethering assays, 3.5 µg of pmirGLO reporter plasmid containing the 3′UTR of Il13 with BoxB were co-transfected with 1 µg of λN-FLAG-HA-Mettl3 expression vector and analyzed 36–48 h post-transfection.

### ELISA

BMMCs were stimulated with IgE and antigen complexes for 6 h, followed by the collection of the cell supernatant. ELISA assay to measure IL-13 concentration was performed using an IL-13 mouse ELISA kit (Invitrogen), following instructions from the manufacturer.

### Passive cutaneous anaphylaxis (PCA)

In vivo PCA experiments were performed as previously described[3]. Ear pinnae of Kit^(Wsh/W-sh) mice lacking mast cells (The Jackson Laboratory)[19] were injected intradermally with $1 \times 10^6$ mast cells. Female mice of 6–8 weeks of age were used. After 2 weeks, ears were sensitized with intradermic injection of 1 µg IgE-anti-DNP antibody. The day after sensitization, mice were challenged with 250 µg HSA-DNP antigen, resuspended in Evans blue dye (10 mg/ml), injected intravenously. Ears were collected and incubated in formamide at 65 °C overnight to retrieve the extravasated blue dye, which correlates with local mast cell activation. Intensity of the dye was measured using a spectrophotometer at optical density 600 nm (OD_{600}).

## Measurement of RNA stability

BMMCs ($2\times10^5$) were seeded in a 48-well plate and incubated with 1 µg/ml IgE–anti-DNP (Sigma) for 30 min followed by stimulation with 0.2 µg/ml HSA-DNP (Sigma) for 30 min. Actinomycin D (10 µg/ml, Sigma) was then added for different time points (15, 30, 60, and 120 min), prior to RNA isolation and RT-qPCR. mRNA decay rate was calculated by non-linear regression curve fitting (one phase decay) using Prism version 9 (GraphPad).

## miCLIP

The miCLIP experiments were performed exactly as previously described[20,54]. Briefly, BMMCs ($6\times10^7$) were either left resting or stimulated with 2 nM PMA (Sigma) and 200 nM ionomycin for 2 h, and total RNA was extracted. The poly-A fraction was purified twice from 3 independent biological replicates using the Dynabeads mRNA purification kit (Ambion) according to the manufacturer's instructions. After fragmentation with an RNA fragmentation reagent (5 min at 70 °C, Invitrogen), the samples were incubated with 6 µg of anti-$m^6$A antibody (Synaptic Systems) in immunoprecipitation buffer (50 mM Tris-HCl, pH 7.4, 100 mM NaCl, 0.05% NP-40) for 4 h at 4 °C, and crosslinked twice with 150 mJ/$cm^2$ of UV light (254 nm) in a UV-Stratalinker. After incubation with 50 µl protein A beads for 1 h at 4 °C, antibody–RNA complexes were washed twice with high-salt buffer (50 mM Tris-HC, pH 7.4, 1 M NaCl, 1 mM EDTA, 1% Igepal CA-630, 0.1% SDS, 0.5% sodium deoxycholate) and twice with polynucleotide kinase (PNK) wash buffer (20 mM Tris-HCl, pH 7.4, 10 mM $MgCl_2$, 0.2% Tween 20). After 3′-end dephosphorylation using PNK beads, the 3′-adapter was ligated with T4 RNA ligase (ligation buffer: 200 mM Tris-HCl, pH 7.8, 40 mM MgCl2, 4 mM DTT). Next, the samples were radioactively labeled, subjected to electrophoresis and transferred onto a Protan BA85 Nitrocellulose Membrane (Whatman). The areas of the membrane containing RNA-crosslinked antibody were excised and the RNA recovered by proteinase K digestion and purification. SuperScript IV reverse transcriptase (Invitrogen) was used for reverse transcription, followed by ligation of the 5′-adapter with T4 RNA ligase (New England Biolabs). After first-strand cDNA synthesis using the Phusion High Fidelity PCR Master Mix (New England Biolabs) and size-selection (ProNex Size-Selective Purification System, Promega), libraries were amplified. Sequencing (50 bp, single-end) was performed on a NextSeq1000 sequencer.

## CLIP read processing

Data processing was performed as previously described[20]. Reads were de-multiplexed based on the experimental barcode and adapter sequences were removed from the read ends (Flexbar v.3.5.0). Unique molecular identifiers (UMIs) were trimmed and added to the read names. Individual samples were mapped to the genome (assembly v.GRCm39) and its annotation (GENCODE release M32) using STAR (v.2.7.6a). Reads were de-duplicated if they had identical UMIs. Significant crosslink sites at single-nucleotide resolution were called using PureCLIP (v.1.3.1) with default parameters, and individual crosslink sites within a distance of 8 bp of each other were merged into wider binding regions (peaks). For assigning a host gene to each PureCLIP peak, transcript annotations were taken from GENCODE release M32, mm39. The metagene plots were generated using the R package Guitar.

## RNA immunoprecipitation-RT-qPCR

BMMCs ($1\times10^7$) were stimulated with 2 nM PMA (Sigma) and 200 nM ionomycin (Sigma) for 2 h. Cells were washed with ice-cold PBS and lysed in lysis buffer (20 mM Tris-HCl, pH7.5, 0.25% NP40, 150 mM NaCl, 1.5 mM $MgCl_2$) supplemented with 1 M DTT, protease inhibitor cocktail (Sigma) and 0.16 U/µl RNase inhibitor (Promega). 1 mg of protein lysate was pre-cleared with 20 µl Dynabeads Protein A (30 mg/ml, Invitrogen) at 4 °C for 1 h on a rotating wheel. 6 µg anti-YTHDF2 (clone EPR20318,

Abcam) or normal rabbit IgG (Cell Signaling Technology) antibody was incubated with 50 µl Dynabeads Protein A (30 mg/ml, Invitrogen) overnight at 4 °C. Pre-cleared lysate was incubated with the antibody-coupled beads at 4 °C for 4 h. Antibody-RNA complexes were recovered on a magnetic stand (Life Technologies) and washed 3 times with ice-cold lysis buffer with RNase inhibitor. RNA was isolated using TRI reagent (MRC) and purified using the Direct-zol RNA mini prep kit (Zymo Research). Reverse transcription was carried out using qScript cDNA SuperMix (Quanta Biosciences) and RT-qPCR was performed using PerfeCTa SYBR Green FastMix (Quanta Bioscience) as described above. Primer sequences are listed in Supplementary Table 2.

## Data analysis

Statistical analysis was performed with Prism version 9 (GraphPad).

## Reporting summary

Further information on research design is available in the Nature Portfolio Reporting Summary linked to this article.

## Data availability

All data generated in this study are provided in the Supplementary Information and Source data files. Sequencing and Nanostring data generated in this study are also available as super-series in the GEO database under accession code GSE228615. Source data are provided with this paper.

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

## Acknowledgements

The authors would like to thank David Jarrossay and Laurent Perez for invaluable technical support, Pamela Nicholson (Next Generation Sequencing Platform, University of Bern) for RNA-seq and Simone Moro for help with the bioinformatic analyses. The HMC-1.1 and 1.2 human mast cell lines were kindly provided by Joseph Butterfield. This work was supported in part by the German Research Foundation grants HE3359/8-1 (#444891219) and HE3359/7-1 (#432656284) (to V.H.), the Fondazione Aldo e Cele Daccò, and the Swiss National Science Foundation grant 310030L_189352 (to S.M.).

## Author contributions

C.L., M.B. and T.I.-K. designed, performed and analyzed experiments. T.I.-K. and V.H. provided essential reagents and protocols. S.M. overviewed the project. C.L. and S.M. wrote the manuscript with input from all authors.

## Competing interests

The authors declare no competing interests.
