## [Peer Review File · Nature Communications]

The mRNA methyltransferase Mettl3 modulates cytokine mRNA stability and limits functional responses in mast cellsReviewers' comments:

Reviewer #1 (Remarks to the Author):

The authors of the manuscript constructed *Mettl3* or *Wtap* deletion mast cells. They observed reduced mast cell proliferation with increased inflammatory cytokine expression in different *in vitro* and *in vivo* models. The phenotype parts of the work (Figures 1-4) are solid. While the authors showed interesting phenotypes of *Mettl3* or *Wtap* deletion in mast cells this work lacks molecular level characterizations.

The authors did not perform experiments to identify m6A-labeled genes in mast cells. They did not report transcripts that are affected by deleting *Mettl3* or *Wtap* through m6A although they checked expression of a few cytokines including IL-13. There is no data to show that IL-13 is m6A-methylated. There is a complete lack of molecular characterization of the deletion. It lacks a comprehensive analysis of how *Mettl3* deficiency affects the whole transcriptome and how that is related to m6A, or perhaps through methylation-independent effects.

No mechanistic work on how m6A might affect cell proliferation and cytokine expression. Are any reader proteins involved and which ones.

I do not feel the current manuscript is up to the standard of a complete story for Nat. Communication.

Reviewer #2 (Remarks to the Author):

In this manuscript, the authors described a new role of METTL3, one catalytic component of m6A methyltransferase complex, in mast cell proliferation and function. Deletion of METTL3 resulted in increased expression of inflammatory cytokines such as IL-13 and TNF, but decreased proliferation capacity of mast cells. This study is interesting and exhibits another example of m6A RNA modification in determining immune cell phenotype and function. However, the related mechanism of METTL3 has not been clearly investigated in current study. Some experimental methods are not reliable and some results are not convincing, lacking solid evidence to support most conclusions.

Major concerns

1. What is the real effect of METTL3 on the proliferation and function of mast cells *in vivo*? Conditional knockout mice with mast cell-specific deletion of *Mettl3* are needed to verify the results in Figure 2 and 4.
2. The authors showed that RNAi or HDR-mediated knockdown of *Mettl3* *in vitro* led to increased expression of inflammatory cytokines but reduced proliferation capacity of mast cells. Since decreased mast cell number may contribute to less cytokine production so as to protect the host from mast cell-mediated inflammation, which effect induced by *Mettl3* deficiency is dominant *in vivo* during IgE-antigen complex stimulation? What is the final outcome of *Mettl3*-deficient mice that suffered from inflammatory diseases such as allergy or asthma?
3. The mechanism by which METTL3 maintains mast cell proliferation was unclear. (1) The authors found that *Mettl3* KO cells had reduced levels of p-STAT5 only in response to IL-3 but not at 0 min, the steady state (Figure 6a), which cannot explain their observation that depletion of METTL3 significantly impaired the basal proliferation capacity of mast cells (Figure 1f and 2d). (2) They did not explain how METTL3 affects STAT5 activation.
4. Figure 3 showed that deletion of WTAP, a component of m6A methyltransferase complex, also impaired mast cell proliferation and accompanied by reduced METTL3 expression. The authors should clarify whether the effect of METTL3 on mast cell proliferation is direct and WTAP-independent. Does

Mettl3 deficiency induce any change of WTAP or other m6A machinery? Which m6A machinery is the most important for maintaining mast cell proliferation?

5. The authors found the upregulation of Mettl3 transcripts in mast cells upon stimulation with IgE and antigen. The protein levels of METTL3, METTL14, WTAP, other m6A methyltransferase complex components, and m6A demethylases in mast cells during the same condition should be showed, with $n \geq 3$ independent experiments. Besides, is there any change of METTL3 protein expression in mast cells stimulated with or without IgE and antigen complexes in vivo?

6. In Figure 5c-g, it would be better to use Mettl3 KO vs. wild-type (non-targeting control template) mast cells. c-Kit KO cells may not be suitable as control for Mettl3 KO cells to identify the target genes of METTL3.

7. In line 438-439, the conclusion "a direct role of Mettl3 catalytic activity in modulating the expression of inflammatory transcripts" lacks evidence, because there was no statistically significant difference between APPA mutant group vs. WT Mettl3 group (Figure 6d right, Figure 6f).

8. Figure 5e-g and 6b-e showed that METTL3 affected the expression and stability of inflammatory transcripts beyond IL-13. A more detailed analysis of METTL3's m6A methylation substrates (m6A-seq of Mettl3 KO vs. WT mast cells) and targeting transcripts (CLIP-seq) could provide insights and confirm whether or not (1) m6A methylation sites exist in 3'UTR of IL-13 transcripts and (2) IL-13 transcripts are directly bound and methylated by Mettl3.

Minor concerns:

1. In Figure 1b and Supplementary Figure 2a, it would be better to show the protein level of METTL3 in mast cells to confirm the efficiency of deletion or overexpression.

2. How or why does IgE-antigen complex treatment decrease METTL3 expression? Discuss more, as this would lead to less mast cell proliferation, does other inflammatory stimulation have the same effect?

3. The authors should provide intact data of flow cytometry analysis and gating strategies for analyzing mast cells. Show mast cell count in bone marrow of unchallenged Mettl3 KO vs. WT mice. Furthermore, the representative figures of flow data related to Figure 1d and 6d should be given.

4. Other immune cells such as basophils also play critical roles or have interaction with mast cells during inflammation and anaphylaxis. Detection of the influence of Mettl3 KO mast cells on other immune cells is suggested.

5. As the authors mentioned that "The functional outcome of m6A is mediated by its recognition by RNA-binding proteins primarily of the YTH family, which upon binding may affect mRNA stability and translation efficiency" in line 65-66. Which m6A reader is responsible for mediating the RNA stability of METTL3's targets?

6. Some results lacked unified statistical analysis, for instance, the control groups were calculated in different ways in Figure 1f vs. Figure 2d. Besides, several results came from only two independent experiments, such as Supplementary Figure 1a, 1b and 4b.

Reviewer #3 (Remarks to the Author):

Leoni et al describe a function for the mRNA methyltransferase Mettl3 in mouse mast cells. They report that Mettl3 negatively regulates the proliferation of mouse mast cells and IgE-mediated

cytokine production and release of early mediators. The regulatory effect by Mettl3 on cytokine production, at least for IL-13, appears to be mediated by methylation of m6A methylation sites in the 3'UTR region of the mRNA that would cause a reduction in the stability of the message. This study may implicate m6A methyltransferase complexes in post-translational regulatory processes of activated mast cells that may help control the inflammatory response.

Overall critique summary:

The findings implicating Mettl3 in post-translational regulation of mast cell responses are novel, but the phenotype is subtle, and its physiological relevance and whether this function for Mettl3 applies to other mast cell models are not addressed. Although the phenotype is consistent using different approaches (knockdown, knockout, overexpression), the choice of some of the controls is surprising or questionable and needs further clarification or study. The authors examine also other phenotypes such as degranulation and proliferation, but only cytokine release is mechanistically linked to direct methylation of the cytokine mRNA. The fact that even immediate responses such as degranulation are affected, suggest a wide range of potential post-translational effects that could alter key components of mast cell specification. The paper could benefit from additional explanations in the logical progression of the manuscripts. It would also be beneficial to move control data or less relevant experiments to supplementary figures.

The specific points are listed below.

Specific comments (in no particular order):

The authors mention that they did the experiments in differentiated mast cells to avoid effects during the differentiation process. Bone-marrow-derived mast cells are however not considered very mature mast cells although they do express FcεRI and c-Kit. Can this type of regulation be generalized to other mast cell types, such as peritoneal mast cells, which are considered more mature and with higher granule content than bone marrow derived? Furthermore, is this also a process in human mast cells?

Since one of the major responses studied in more detail is cytokine regulation, a relevant model of inflammation and dependent on mast cell cytokines would be needed to determine the physiological importance of the observations in vitro. The passive cutaneous anaphylaxis experiment is mostly used as an in vivo model for mast cell degranulation, which does not seem to be the main focus of the paper (as it was not explored using all the experimental approaches like was done the cytokine production, or was it linked to methylation of specific targets).

Cytokine released into the media should also be tested by ELISA to better gauge how the subtle differences in the percentage of cells expressing the cytokine reflects on actual protein release.

Did the effect of Mettl3 knockdown or KO on proliferation was also accompanied by increased cell death?

Did knockdown of Mettl3 by siRNA or CRISPR cause changes in c-Kit expression? Kit is a major receptor for mast cell proliferation and survival and can signal through Stat5. It should be important to show the expression of Kit and FcεRI after knockdown or knockout of Mettl3. In Supplementary table 3, it appears that the expression of Kit in the c-Kit KO cells is similar to that in Mettl3 KO cells. This is also confusing because c-Kit KO has high raw counts for the nanostring.

Related to the latter, why was c-Kit depletion by CRISPR a control for Mettl3 KO? Was the transcriptome of c-Kit KO using CRISPR any different to normal cells? C-Kit is an essential receptor for mast cell proliferation, survival and function, and even in the absence of SCF, serum may contain enough amounts of SCF to affect these functions via c-Kit. Removal of Kit may alter the behavior/make-up of the cells and therefore may not be the best control. The authors should confirm

that mast cell specific gene signature (for example as found in doi:10.1038/nr3445) is not changed between c-kit KO and normal mast cells.

A potentially better or additional control for the Mettl3 study would be Mettl14 KO, or even unaltered cells. Did knockdown of Mettl14 have a functional effect on cytokine production?

Although it is understandable why the troubleshooting of the KO technique was established using c-Kit as a model target, all the explanations etc. in the text and figures related to this are quite distracting from the main story and could be moved to the method and supplementary section because they don't seem relevant to the paper except from the methodological standpoint.

Are induction and effects of Mettl3 specific to IgE-mediated responses? Other stimulus for mast cells such as IL-33 (alone or together with FcεRI activation) produce more robust cytokine responses and longer lasting than Ag alone which may relate in part to message stability.

Supplementary Figure 1a and b are relevant and should be within the main paper, while Figure 1a is a control experiment that should be instead in supplemental. The efficiency of transfection in this control experiment, although useful, does not imply equal efficiency for the construct from Mettl3. In addition, the whole blots (instead of or in addition to the cropped images) for Mettl3 should be provided. Also, a western blot showing depletion at the protein level when using knockdown of Mettl3 should be included to demonstrate specificity of the bands and efficiency of knockdown.

Method descriptions could use in general more detail or explanations. For instance (but not restricted to), in the flow cytometry section, what was the percentage of FcεRI/c-kit double positive cells in the cultures? Were live cells and FcεRI/c-kit double positive gated for the cytokine measurements? Were monensin/brefeldin/or golgi stop used for cytokine measurements? If so, please describe. Another example is the T7 endonuclease assay, where the expected sizes and explanations of the products are not described, so Figure 2 C cannot be interpreted. Another instance is the BoxB constructs are not described in Methods

Related to Figure 1B and others, what are the average Ct values for Mettl3 before and after knockdown? This is important to have a better impression of how abundant this message is.

Figure 2A should be supplementary

Figure 2B, first two quadrants have no labels no RNP and Mettl3 RNP?

The differences described are relative mild, but the authors use language that is too strong such as "profound", "crucial". Should tone it down

Please revise the use of uppercase or lowercase letters and italics for Mettl3 throughout to use the proper designation (mouse vs human; gene vs protein).

Point-by-point reply to the Reviewers' comments:

We thank the Reviewers for providing constructive comments that allowed us to extend our study and further strengthen our conclusions. Major additions that are now included in the manuscript (in track changes in the main text) are summarized below, followed by a detailed point-by-point response. Briefly:

- 1) We performed m⁶A-CLIP-seq experiments to identify transcripts that are methylated specifically in mast cells, either resting or stimulated, at a near-nucleotide resolution. We found that upon activation, the *Il13* transcript is prominently methylated in this cell type, greatly strengthening our previous conclusions (**new Figure 5**).
- 2) We expanded our previous findings by performing RNA-seq analyses of mast cells wild-type or Mettl3-KO, allowing us to comprehensively describe how Mettl3 deficiency affects the mast cell transcriptome (**new Figure 5**).
- 3) We further validated the importance of different components of the m⁶A methyltransferase complex in modulating mast cell proliferation and function by deleting Mettl14, revealing that its ablation phenocopied Mettl3 and Wtap deletions (**new Figure 4**).
- 4) Using human mast cell lines and *ex vivo*-derived peritoneal mast cells, we confirmed that the expression dynamics and basic functions of Mettl3 are conserved across different experimental systems (**new Supplementary Figure 2d and new Figure 1b**).

Overall, these data further extend and strengthen our previous conclusions, and we are grateful to the Reviewers for their suggestions.

Reviewer #1 (Remarks to the Author):

The authors of the manuscript constructed Mettl3 or Wtap deletion mast cells. They observed reduced mast cell proliferation with increased inflammatory cytokine expression in different in vitro and in vivo models. The phenotype parts of the work (Figures 1-4) are solid. While the authors showed interesting phenotypes of Mettl3 or Wtap deletion in mast cells this work lack molecular level characterizations.

The authors did not perform experiments to identify m6A-labeled genes in mast cells. They did not report transcripts that are affected by deleting Mettl3 or Wtap through m6A although they checked expression of a few cytokines including IL-13. There is no data to show that IL-13 is m6A-methylated. There is a complete lack of molecular characterization of the deletion. It lacks a comprehensive analysis of how Mettl3 deficiency affects the whole transcriptome and how that is related to m6A, or perhaps through methylation-independent effects.

No mechanistic work on how m6A might affect cell proliferation and cytokine expression. Are any reader proteins involved and which ones.

I do not feel the current manuscript is up to the standard of a complete story for Nat. Communication.

We now extended the molecular characterization of Mettl3-deleted mast cells by performing both m⁶A-CLIP-seq and RNA-seq, as requested. Specifically:

- 1) **m⁶A-CLIP-seq**: We performed m⁶A-CLIP-seq experiments using resting and activated mast cells, to determine which transcripts are specifically methylated in this cell type in these conditions. We found that

in resting cells a number of transcripts were methylated and upregulated upon *Mettl3* deletion (**new Figure 5c-e**).

Figure 5e. Transcripts methylated (by m⁶A-CLIP-seq) and differentially expressed (by RNA-seq) in *Mettl3* knock-out mast cells.

Upon activation, the *Il13* transcript (among other inflammatory transcripts) was prominently methylated in mast cells, greatly strengthening our previous conclusions (**new Figure 5f**).

Figure 5f. The 3'UTR of the *Il13* transcript is m⁶A-methylated in activated mast cells.

2) **RNA-seq:** We performed whole transcriptome analysis of wild-type and *Mettl3*-KO mast cells, as requested, to further extend our previous findings. Consistent with our previous analysis using Nanostring digital profiling, we found that in resting mast cells more genes were upregulated than downregulated in absence of *Mettl3*. Some of these genes were associated to cell proliferation and intracellular trafficking, potentially explaining at least some of the phenotypes observe in mast cells lacking *Mettl3* (**new Figure 5a-b**).

Figure 5a. RNA-seq analysis of wild-type and Mettl3-knock-out mast cells.

3) To address the part of the question related to possible **methylation-independent effects**, we performed the deletion of other components of the methyltransferase complex. We found that in most cases, deletion of one component also affected the expression of other components of the complex (**Figure 3b**, **new Figure 4f** and **new Supplementary Figure 4**), including Mettl3 itself, making it impossible to fully distinguish the activities of the complex that are linked specifically to its catalytic activities to those that are not. However, we found that deletion of Mettl14 phenocopied the deletion of Mettl3 (**new Figure 4**), further strengthening our conclusions and highlighting the importance of the entire complex for these mast cell phenotypes.

Figure 4b-d. Mettl14 deletion phenocopies Mettl3 ablation in mast cells, leading to reduced proliferation (b), survival (c) and increased IL-13 production (d).

Supplementary Figure 4. Deletion of Mettl3 affects the expression of other components of the methyltransferase complex. a) Western blot; b) intracellular staining.

4) Regarding the involvement of m^6A reader proteins, first we measured the expression of the transcripts encoding the different YTH proteins in resting and stimulated mast cells. Except for Ythdc2, that was not expressed (new **Supplementary Figure 6b**), all other transcripts were expressed at comparable levels and remained unchanged upon mast cell stimulation with IgE and antigen complexes (new **Supplementary Figure 6c**). Since Ythdc1 acts in the cell nucleus¹, we focused our attention on the cytoplasmic readers of the Ythdf family. Given the similar expression levels and the functional redundancy of these proteins¹, we focused on Ythdf2 for technical reasons and antibody availability, and we found that Ythdf2 has the ability to bind the *Il13* transcript in stimulated mast cells (new **Supplementary Figure 6d**).

Supplementary Figure 6d. RIP-RT-qPCR for the *Il13* 3'UTR, using an anti-YTHDF2 antibody. Mettl3-depleted cells were also used as control (white bars). R1 and R2 refer to two different regions amplified by RT-qPCR in the *Il13* 3'UTR.

Overall, we now provided new data revealing the identity of m⁶A-labelled transcripts in mast cells, and we found that these included *III3*. We also provided a comprehensive analysis of how Mettl3 deficiency affects the whole transcriptome, and we showed that at this stage it is not possible to completely separate direct and indirect effects of the methyltransferase complex, since removal of one component also affects the expression of other components. However, we now show that removal of Mettl14 phenocopied Mettl3 deletion, and we provide mechanistic details showing that increased *III3* stability in absence of Mettl3 is causally linked to m⁶A methylation in the *III3* 3'UTR, requiring both the DRACH motifs in the 3'UTR and the catalytic activity of Mettl3 (**Figure 8**).

Reviewer #2 (Remarks to the Author):

In this manuscript, the authors described a new role of METTL3, one catalytic component of m6A methyltransferase complex, in mast cell proliferation and function. Deletion of METTL3 resulted in increased expression of inflammatory cytokines such as IL-13 and TNF, but decreased proliferation capacity of mast cells. This study is interesting and exhibits another example of m6A RNA modification in determining immune cell phenotype and function. However, the related mechanism of METTL3 have not been clearly investigated in current study. Some experimental methods are not reliable and some results are not convincing, lacking solid evidence to support most conclusions.

Major concerns

1. What is the real effect of METTL3 on the proliferation and function of mast cells in vivo? Conditional knockout mice with mast cell-specific deletion of *Mettl3* are needed to verify the results in Figure 2 and 4.

We appreciate the reviewer suggestion. Since we observed the same cell-intrinsic effects of *Mettl3* using a variety of experimental approaches (siRNAs, standard CRISPR-Cas9 knock-out and gene replacement by HDR), both *in vitro* and in adoptive transfer experiments *in vivo*, we have no indication at this stage that including additional models would provide radically different results, at least regarding cell-intrinsic effects.

However, to independently validate the results in Figure 2 and 4 (now Figure 2 and 6), we repeated our experiments using mast cells lacking *Mettl14*, another central component of the methyltransferase complex (**new Figure 4**). We found that the absence of *Mettl14* phenocopied the *Mettl3* deletion, both in terms of mast cell phenotypes (reduced proliferation and survival, increased cytokine production) and in the destabilization of other components of the methyltransferase complex. Overall, our results were independently validated in a large number of different experimental systems.

2. The authors showed that RNAi or HDR-mediated knockdown of *Mettl3* *in vitro* led to increased expression of inflammatory cytokines but reduced proliferation capacity of mast cells. Since decreased mast cell number may contribute to less cytokine production so as to protect the host from mast cell-mediated inflammation, which effect induced by *Mettl3* deficiency is dominant *in vivo* during IgE-antigen

complex stimulation? What is the final outcome of *Mettl3*-deficient mice that suffered from inflammatory diseases such as allergy or asthma?

Actually, despite the reduced viability and proliferation capacity of cells lacking *Mettl3*, we found that mast cells lacking *Mettl3* induced exacerbated *in vivo* inflammation compared to wild-type cells (**Figure 2h**), suggesting that the increased release of inflammatory mediators and cytokines is the dominant effect *in vivo* during IgE-antigen complex stimulation.

As for the use of other models of allergy or asthma, we appreciate the reviewer suggestion, and we agree that it would be interesting to initiate a new study including the generation of various conditional *Mettl3* alleles and assessing their impact on a variety of disease models. However, the role of *Mettl3* in inflammatory disease was not the focus of our current work, where we instead elected to perform an in-depth analysis of the cell-intrinsic effects of *Mettl3* ablation in mast cells.

3. The mechanism by which *METTL3* maintains mast cell proliferation was unclear. (1) The authors found that *Mettl3* KO cells had reduced levels of p-STAT5 only in response to IL-3 but not at 0 min, the steady state (Figure 6a), which cannot explain their observation that depletion of *METTL3* significantly impaired the basal proliferation capacity of mast cells (Figure 1f and 2d). (2) They did not explain how *METTL3* affects STAT5 activation.

We apologize for the unclear description of our experiment. The ones mentioned by the reviewer are different experimental setups that cannot be directly compared: 0 min of stimulation is not representative of the steady-state, but rather of the state of the cells after 18h of IL-3 withdrawal. There is no Stat5 signaling (or proliferation) in these conditions. Basal proliferation (steady state) only occurs with saturating amounts of IL-3, normally present in the cultures at all times². Therefore, the reduced ability of the cells to signal through the IL-3 receptor in the absence of *Mettl3* may explain, at least in part, the reduced proliferation.

By performing m⁶A-CLIP-seq, we found that transcripts encoding regulators of cytokine signaling are methylated in these cells (new **Figure 5**).

Figure 5f. Transcripts encoding regulators of cytokine signaling are methylated in mast cells, as detected by m⁶A-CLIP-seq.

However, despite being methylated, the abundance of these transcripts did not change in a significant manner upon *Mettl3* deletion, a finding that was already described in the previous version of our manuscript and that was once more confirmed by our new RNA-seq experiment (new **Figure 5**). The reason behind this apparent disconnect between methylation and expression level for these transcripts remains to be investigated, but it may be linked to a balance between transcription rate and stability of the transcripts themselves. Since our new transcriptome analysis revealed that other transcripts encoding regulators of cell proliferation and cytoskeletal reorganization are significantly differentially expressed by

Mettl3-KO cells, we now removed the previous experiments related to Stat5, that still lacked a comprehensive mechanistic understanding. We thank the Reviewer for pointing this out.

4. Figure 3 showed that deletion of WTAP, a component of m6A methyltransferase complex, also impaired mast cell proliferation and accompanied by reduced METTL3 expression. The authors should clarify whether the effect of METTL3 on mast cell proliferation is direct and WTAP-independent. Does Mettl3 deficiency induce any change of WTAP or other m6A machinery? Which m6A machinery is the most important for maintaining mast cell proliferation?

We thank the Reviewer for allowing us to clarify this important point. The effect of Mettl3 on mast cell proliferation is very unlikely to be Wtap-independent, since both proteins are integral parts of the m⁶A machinery complex, and the complex in its entirety is required to maintain mast cell proliferation. Indeed, a recent cryo-EM structure of human m⁶A writer complexes highlighted how the catalytic and non-catalytic components of the complex assemble mainly through WTAP and METTL3 interactions^{3,4}. Most likely, while Mettl3 has enzymatic capacity and is required for methylation, the accessory proteins are required for RNA binding and for the interaction with other mRNA machineries, including splicing, tailing, etc.

As for the relative importance of the different components of the m⁶A machinery in modulating mast cell proliferation, to address this question, we measured the stability of Wtap and other components of the complex (Mettl14, Virma) in cells lacking Mettl3. We found that deletion of Mettl3 affected the expression of other components of the methyltransferase complex, further highlighting their obligatory interaction. Specifically, Mettl14 expression was diminished, while Wtap levels were increased (**new Suppl. Figure 4**). Virma expression was also modestly increased, albeit not significantly. These results indicate that it is not possible at present to fully separate and isolate the role of each individual component of the m⁶A writer complex.

Supplementary Figure 4. Deletion of *Mettl3* affects the expression of other components of the methyltransferase complex. a) Western blot; b) intracellular staining.

Next, we deleted the *Mettl14* gene to determine whether its ablation phenocopied *Mettl3* and *Wtap* deletion in mast cells. Concordant with the data above, we found that *Mettl14* deletion affected the expression of *Mettl3* (new **Figure 4e-f**) and phenocopied the deletion of *Mettl3*, including reduced viability and proliferation, and increased cytokine production (new **Figure 4**). Overall, our results reveal that the methyltransferase complex as a whole is required for proper mast cell regulation. We thank the reviewer for allowing us to clarify these important aspects.

Figure 4f. *Mettl14* deletion destabilizes *Mettl3* expression.

5. The authors found the upregulation of *Mettl3* transcripts in mast cells upon stimulation with IgE and antigen. The protein levels of *METTL3*, *METTL14*, *WTAP*, other m6A methyltransferase complex components, and m6A demethylases in mast cells during the same condition should be showed, with $n \geq 3$ independent experiments. Besides, is there any change of *METTL3* protein expression in mast cells stimulated with or without IgE and antigen complexes in vivo?

We performed the requested expression analyses for different components of the methyltransferase complex, including *Mettl3*, *Wtap* and *Virma*, by stimulating mast cells with either IgE and antigen complexes or PMA and ionomycin. A western blot analysis showing *Mettl3* protein upregulation upon IgE and antigen stimulation is now shown in the new **Figure 1a**.

Figure 1a. *Mettl3* is induced upon stimulation of mast cells with IgE and antigen complexes.

We also analyzed the expression of Wtap and Virma by intracellular staining. We found that they were both modestly upregulated (**Figure for Reviewer** below). We modified the description of our results in the text to tone down the importance of Mettl3 upregulation upon stimulation.

Figure for Reviewer. Virma (left) and Wtap (right) are only modestly induced by stimulation of mast cells with IgE and antigen complexes or PMA and ionomycin. Each dot represents one independent experiments. Mean \pm SEM, paired t-test.

Finally, we also found that Mettl3 expression was induced upon stimulation of peritoneal cavity-derived mast cells (new **Figure 1b**), pointing towards similar mechanisms of regulation also *in vivo*. Overall, these new data further confirm and strengthen our previous findings.

Figure 1b. Mettl3 expression is increased upon stimulation of peritoneal-derived mast cells (PMCs) with IgE and antigen. *p=0.0148

6. In Figure 5c-g, it would be better to use Mettl3 KO vs. wild-type (non-targeting control template) mast cells. c-Kit KO cells may not be suitable as control for Mettl3 KO cells to identify the target genes of METTL3.

We now repeated the requested experiment by comparing wild-type and Mettl3-KO cells. Cells were subjected to RNA-seq and the new data are now included in the new **Figure 5a-b**.

Figure 5a. RNA-seq analysis of wild-type and Mettl3-knock-out mast cells.

7. In line 438-439, the conclusion “a direct role of *Mettl3* catalytic activity in modulating the expression of inflammatory transcripts” lacks evidence, because there was no statistically significant difference between APPA mutant group vs. WT *Mettl3* group (Figure 6d right, Figure 6f).

For Figure 6f (luciferase assay, now **Figure 8f**), we now performed a few additional experiments, which improved the statistics (**p value = 0.0045). We thank the Reviewer for allowing us to further strengthen our conclusions.

Figure 8f. *Mettl3* catalytic activity is required to modulate expression of a luciferase reporter containing the *Il13* 3'UTR. Note that the complementation of cell lines lacking endogenous METTL3 (left) is required to reveal the differences between WT and APPA *Mettl3*.

We also performed new experiments to improve the statistics of the experiments with lentiviral transduction of wild-type mast cells with WT or APPA *Mettl3*. As already previously discussed in the manuscript, it is very difficult to observe any effect of the APPA mutant in wild-type cells, which already express high levels of endogenous wild-type *Mettl3*. To be able to perform luciferase assay and detect any effect that would be otherwise masked by the endogenous protein, we had to generate METTL3-KO HEK293 cells (as described in **Figure 8f**) to be then complemented with either WT or APPA *Mettl3*. This experiment was not possible to perform in primary mast cells for technical reasons. Therefore, while overexpression of *Mettl3* led to the robust downregulation of cytokine expression, overexpression of the APPA mutant in wild-type cells led to high variability that could not possibly achieve statistical significance, most likely because of competition with the highly expressed endogenous wild-type *Mettl3*. Although all our experiments point towards a clear requirement for both *Mettl3* expression and catalytic activity in modulating the expression of inflammatory transcripts, due to the variability of this one particular experiment we toned down our conclusions as requested.

8. *Figure 5e-g and 6b-e* showed that METTL3 affected the expression and stability of inflammatory transcripts beyond *IL-13*. A more detailed analysis of METTL3's m⁶A methylation substrates (m⁶A-seq of *Mettl3* KO vs. WT mast cells) and targeting transcripts (CLIP-seq) could provide insights and confirm whether or not (1) m⁶A methylation sites exist in 3'UTR of *IL-13* transcripts and (2) *IL-13* transcripts are directly bound and methylated by *Mettl3*.

To address these important points, we now performed the m⁶A-CLIP-seq experiments as requested, which allowed us to establish that indeed the 3'UTR of the *Il13* transcript is methylated in stimulated mast cells (new **Figure 5**). We also provide mechanistic details showing that increased *Il13* stability in absence of *Mettl3* is causally linked to m⁶A methylation in the *Il13* 3'UTR, requiring both the DRACH motifs in the 3'UTR and the catalytic activity of *Mettl3* (**Figure 8**).

Figure 5f. The 3'UTR of the *I13* transcript is m⁶A-methylated in activated mast cells.

Minor concerns:

1. In Figure 1b and Supplementary Figure 2a, it would be better to show the protein level of METTL3 in mast cells to confirm the efficiency of deletion or overexpression.

We now included a western blot to show depletion of Mettl3 by RNAi (new **Supplementary Figure 1e**). The overexpression of Mettl3 at a protein level is shown by intracellular staining in **Figure 8b**.

(Left) Suppl. Figure 1e. Depletion of Mettl3 in siRNA transfected mast cells. **(Right) Figure 8b.** Overexpression of Mettl3 in mast cells, measured by intracellular staining.

2. How or why does IgE-antigen complex treatment decrease METTL3 expression? Discuss more, as this would lead to less mast cell proliferation, does other inflammatory stimulation have the same effect?

Actually, we showed that mast cell stimulation increases Mettl3 expression. To determine whether this induction is strictly signal-dependent, or it occurs in response to a variety of inflammatory signals, we measured Mettl3 expression also in response to LPS, IL-33 or SCF stimulation. By intracellular staining, we found no evidence of a significant induction in Mettl3 expression by other stimuli (**Figure for Reviewer** below). The mechanism regulating Mettl3 expression in response to IgE crosslinking remains to be understood.

Figure for Reviewer. No evidence of a significant induction in Mettl3 expression by stimuli other than IgE and antigen. Each dot represents one independent experiment. Mean \pm SEM. Paired t-test. 10 ng/mL IL-33 (Biologend), 10 ng/mL SCF (Peprotech), 1 μ g/ml LPS (Invivogen).

3. The authors should provide intact data of flow cytometry analysis and gating strategies for analyzing mast cells. Show mast cell count in bone marrow of unchallenged *Mettl3* KO vs. *WT* mice. Furthermore, the representative figures of flow data related to Figure 1d and 6d should be given.

We now included FACS plot analyses and gating strategies in the Supplementary Information. Representative FACS plot related to Figure 1d (now Figure 1e) were provided in **Supplementary Figure 2b**. Additional representative flow data for Figure 6d (now Figure 8c) are now provided in **Supplementary Figure 6a**). Please note that nowhere in this study we used *Mettl3*-KO mice, that are embryonically lethal⁵. We also do not have or have access to any conditional *Mettl3*^{-/-} allele. Finally, please note that there are no or extremely rare mast cells in the bone marrow of mice, which contains only progenitors.

4. Other immune cells such as basophils also play critical roles or have interaction with mast cells during inflammation and anaphylaxis. Detection of the influence of *Mettl3* KO mast cells on other immune cells is suggested.

We thank the Reviewer for this suggestion. However, since any finding about the role of *Mettl3* in other immune cell types would have no impact on our conclusions about the cell-intrinsic role of *Mettl3* in mast cells, we decided not to pursue this point.

5. As the authors mentioned that “The functional outcome of m6A is mediated by its recognition by RNA-binding proteins primarily of the YTH family, which upon binding may affect mRNA stability and translation efficiency” in line 65-66. Which m6A reader is responsible for mediating the RNA stability of *METTL3*'s targets ?

We measured the expression of the transcripts encoding the different YTH proteins in resting and stimulated mast cells (new **Supplementary Figure 6b-c**). We then focused our attention on the cytoplasmic readers of the Ythdf family. Given the similar expression levels and the functional redundancy of these proteins¹, we focused on Ythdf2 for technical reasons and antibody availability. We found that Ythdf2 can be detected to bind the *Il13* transcript in stimulated mast cells (new **Supplementary Figure 6d**), although whether other RNA-binding proteins bind the *Il13* 3'UTR in a m⁶A-dependent or -independent manner remains to be fully understood and will be the subject of future work.

Supplementary Figure 6d. RIP-RT-qPCR for the *Il13* 3'UTR, using an anti-YTHDF2 antibody. *Mettl3*-depleted cells were also used as control (white bars). R1 and R2 refer to two different regions amplified by RT-qPCR in the *Il13* 3'UTR.

6. Some results lacked unified statistical analysis, for instance, the control groups were calculated in different ways in Figure 1f vs. Figure 2d. Besides, several results came from only two independent experiments, such as Supplementary Figure 1a, 1b and 4b.

We thank the Reviewer for pointing this out. We now included new western blots and analyses, confirming and extending our previous results as detailed above. We also unified the representation of the data as recommended, providing in both figures the normalized data (new **Figure 1g**).

Reviewer #3 (Remarks to the Author):

Leoni et al describe a function for the mRNA methyltransferase *Mettl3* in mouse mast cells. They report that *Mettl3* negatively regulates the proliferation of mouse mast cells and IgE-mediated cytokine production and release of early mediators. The regulatory effect by *Mettl3* on cytokine production, at least for IL-13, appears to be mediated by methylation of m6A methylation sites in the 3'UTR region of the mRNA that would cause a reduction in the stability of the message. This study may implicate m6A methyltransferase complexes in post-translational regulatory processes of activated mast cells that may help control the inflammatory response.

Overall critique summary:

The findings implicating *Mettl3* in post-translational regulation of mast cell responses are novel, but the phenotype is subtle, and its physiological relevance and whether this function for *Mettl3* applies to other mast cell models are not addressed. Although the phenotype is consistent using different approaches (knockdown, knockout, overexpression), the choice of some of the controls is surprising or questionable and needs further clarification or study. The authors examine also other phenotypes such as degranulation and proliferation, but only cytokine release is mechanistically linked to direct methylation of the cytokine mRNA. The fact that even immediate responses such as degranulation are affected, suggest a wide range of potential post-translational effects that could alter key components of mast cell specification. The paper could benefit from additional explanations in the logical progression of the manuscripts. It would also be beneficial to move control data or less relevant experiments to supplementary figures.

The specific points are listed below.

Specific comments (in no particular order):

The authors mention that they did the experiments in differentiated mast cells to avoid effects during the differentiation process. Bone-marrow-derived mast cells are however not considered very mature mast cells although they do express *FcεRI* and *c-Kit*. Can this type of regulation be generalized to other mast cell types, such as peritoneal mast cells, which are considered more mature and with higher granule content than bone marrow derived? Furthermore, is this also a process in human mast cells?

We thank the Reviewer for pointing this out. First, we measured *Mettl3* expression in *ex-vivo*-derived peritoneal mast cells (PMCs). Consistent with our previous findings, stimulation of PMCs with IgE and antigen complexes increased *Mettl3* protein expression, measured by intracellular staining (new **Figure 1b**), suggesting that a similar mechanism is at play in peritoneal cells.

Figure 1b. *Mettl3* expression is increased upon stimulation of peritoneal-derived mast cells (PMCs) with IgE and antigen.

Next, we repeated our analyses using bone marrow-derived mast cells cultured in the presence SCF, which leads to a more differentiated cell phenotype. We found that upon *Mettl3* deletion most of the phenotypes were maintained, including increased cytokine expression and reduced viability. Interestingly, SCF treatment was however able to compensate the proliferation defect, with a mechanism that remains to be understood. These data are now presented in the new **Supplementary Figure 3d-f**.

Finally, we deleted *METTL3* in the human mast cell lines HMC-1.1 and 1.2, which led to reduced proliferation and increased cytokine production by these cells (we could only measure TNF, since IL-13 was not expressed by these cells), pointing towards an evolutionary conserved mechanism (new **Supplementary Figure 2d**).

Suppl. Figure 2d. *METTL3* ablation in human mast cell lines reduces cell proliferation and increases TNF production.

Overall, we confirmed that the expression dynamics and basic functions of *Mettl3* are conserved across different experimental systems. We thank the Reviewer for allowing us to further strengthen our data.

Since one of the major responses studied in more detail is cytokine regulation, a relevant model of inflammation and dependent on mast cell cytokines would be needed to determine the physiological importance of the observations in vitro. The passive cutaneous anaphylaxis experiment is mostly used as an in vivo model for mast cell degranulation, which does not seem to be the main focus of the paper (as it was not explored using all the experimental approaches like was done the cytokine production, or was it linked to methylation of specific targets).

Cytokine released into the media should also be tested by ELISA to better gauge how the subtle differences in the percentage of cells expressing the cytokine reflects on actual protein release.

The phenotypic effects of deleting *Mettl3* (and, as we show in our revised manuscript, other components of the methyltransferase complex) are multifaceted and undoubtedly involve a number of direct and indirect effects. For now, we were able to establish that such phenotypes are strictly linked to the expression of the methyltransferase complex, and that at least one of them (namely IL-13 production) is mechanistically linked to altered mRNA stability. To extend our description of cytokine expression upon *Mettl3* deletion, we performed an ELISA assay as recommended, showing increased IL-13 production (new **Figure 2d**). Furthermore, we performed RNA-seq analyses of wild-type and *Mettl3*-KO cells, which

revealed that the dysregulated genes were linked to cell proliferation, and also to intracellular trafficking, potentially contributing to the different phenotypes observed (new **Figure 5**).

Figure 2d. IL-13 secretion in the supernatant of wild-type and Mettl3 KO mast cells, measured by ELISA.

Did the effect of Mettl3 knockdown or KO on proliferation was also accompanied by increased cell death?

We thank you the reviewer for pointing this out. Indeed, Mettl3 deletion also resulted in modestly but significantly reduced cell viability. These data are now reported in the new **Figure 2g**.

Figure 2g. Deletion of Mettl3 reduces mast cell viability.

Did knockdown of Mettl3 by siRNA or CRISPR cause changes in c-Kit expression? Kit is a major receptor for mast cell proliferation and survival and can signal through Stat5. It should be important to show the expression of Kit and Fcεr1 after knockdown or knockout of Mettl3. In Supplementary table 3, it appears that the expression of Kit in the c-Kit KO cells is similar to that in Mettl3 KO cells. This is also confusing because c-Kit KO has high raw counts for the nanostring.

We thank the reviewer for allowing us to clarify this point. First, c-Kit surface expression was unaffected by Mettl3 (and Mettl14) ablation, as shown by FACS staining (new **Supplementary Figure 3c**).

Suppl. Figure 3c. Comparable c-Kit expression in wild-type and Mettl3 knock-out mast cells.

As for our Nanostring data, they show that the expression of the *Kit* mRNA is apparently unchanged simply because the probes to detect mRNA expression in the Nanostring panel are located very far away from the deletion (see **Figure for Reviewer** below). Indeed, successful Cas9 editing creates mutations, insertion and deletions, but it does not necessarily impact RNA expression directly. In fact, we found that in the conditions used for c-Kit deletion, mRNA expression was unaffected (at least when a probe located in the 3'-UTR of the gene was used), while surface protein expression was completely ablated (**Figure 6c**).

Figure for Reviewer. Left: location of the Nanostring probe compared to the CRISPR-Cas9 deletion in the *Kit* gene. Right: abrogated protein expression in *Kit*-deleted cells.

Related to the latter, why was c-Kit depletion by CRISPR a control for Mettl3 KO? Was the transcriptome of c-Kit KO using CRISPR any different to normal cells? C-Kit is an essential receptor for mast cell proliferation, survival and function, and even in the absence of SCF, serum may contain enough amounts of SCF to affect these functions via c-Kit. Removal of Kit may alter the behavior/make-up of the cells and therefore may not be the best control. The authors should confirm that mast cell specific gene signature (for example as found in doi:10.1038/ni.3445) is not changed between c-kit KO and normal mast cells. A potentially better or additional control for the Mettl3 study would be Mettl14 KO, or even unaltered cells. Did knockdown of Mettl14 have a functional effect on cytokine production?

We thank the reviewer for allowing us to clarify this important point. First, we observed the same effect of Mettl3 depletion/ deletion on the mast cell phenotype in a variety of experimental conditions, most of which did not involve the *Kit* gene as a control. Specifically, similar effects on cytokine expression were observed in Mettl3 KO cells compared to mock transfected controls, and in Mettl3-depleted cells using siRNAs compared to non-targeting controls. We are therefore very confident that the observed effects were not simply due to altered c-Kit expression. However, we also performed the requested knock-out of the *Mettl14* gene, which showed increased cytokine expression, consistent with our previous data (new **Figure 4**).

Figure 4b-d. Mettl14 deletion phenocopies Mettl3 ablation in mast cells, leading to reduced proliferation (b), survival (c) and increased IL-13 production (d).

Finally, we repeated the transcriptome analysis of Mettl3 knock-out cells in comparison with scramble-control wild-type cells (rather than *Kit* knock-out cells). The results are now shown in the new **Figure 5a-b**.

Figure 5a. RNA-seq analysis of wild-type and Mettl3-knock-out mast cells.

Although it is understandable why the troubleshooting of the KO technique was established using c-Kit as a model target, all the explanations etc. in the text and figures related to this are quite distracting from the main story and could be moved to the method and supplementary section because they don't seem relevant to the paper except from the methodological standpoint.

We appreciate the Reviewer's point of view, but we think that our study will be relevant for the community also from the methodological angle, and we therefore wish to provide adequate visibility to this important aspect of our work.

Are induction and effects of Mettl3 specific to IgE-mediated responses? Other stimulus for mast cells such as IL-33 (alone or together with FcεR1 activation) produce more robust cytokine responses and longer lasting than Ag alone which may relate in part to message stability.

Although IL-33 was unable to induce Mettl3 expression, it indeed greatly potentiated cytokine responses to IgE crosslinking. Nevertheless, deletion of Mettl3 still augmented the capacity of mast cells to produce cytokines even upon stimulation with IL-33 plus IgE and antigen (new **Supplementary Figure 3g**).

Supplementary Figure 3g. Increased cytokine production in mast cells lacking Mettl3 and stimulated with IgE and antigen, with or without IL-33.

Supplementary Figure 1a and b are relevant and should be within the main paper, while Figure 1a is a control experiment that should be instead in supplemental. The efficiency of transfection in this control experiment, although useful, does not imply equal efficiency for the construct from Mettl3. In addition, the whole blots (instead of or in addition to the cropped images) for Mettl3 should be provided. Also, a western blot showing depletion at the protein level when using knockdown of Mettl3 should be included to demonstrate specificity of the bands and efficiency of knockdown.

We reorganized the panels as recommended by the Reviewer. The whole blots are provided in the **Source Data Files**. A western blot showing depletion of Mettl3 by RNAi is now shown in the new **Supplementary Figure 1e**.

Supplementary Figure 1e. Depletion of Mettl3 in siRNA transfected mast cells.

Method descriptions could use in general more detail or explanations. For instance (but not restricted to), in the flow cytometry section, what was the percentage of FcεRI/c-kit double positive cells in the cultures? Were live cells and FcεRI/c-kit double positive gated for the cytokine measurements? Were monensin/brefeldin/or golgi stop used for cytokine measurements? If so, please describe. Another example is the T7 endonuclease assay, where the expected sizes and explanations of the products are not

described, so Figure 2 C cannot be interpreted. Another instance is the BoxB constructs are not described in Methods

We apologize for the lack of details in some paragraphs. We now improved these descriptions throughout the methods and/ or the results sections. Specific for the T7, we now included a schematic representation of the expected results both in the main figure (**Figure 2c**) and a detailed schematic in **Supplementary Figure 3b**.

Supplementary Figure 3b. Schematic representation of the *Mettl3* locus with indicated the locations of the gRNAs used in this study, the length of the PCR products used for T7 endonuclease I assay in Figure 2c and the expected size of the bands in this assay.

Related to Figure 1B and others, what are the average Ct values for Mettl3 before and after knockdown? This is important to have a better impression of how abundant this message is.

The mRNA encoding *Mettl3* is rather abundant in these cells, around 24 Cts, which is close to the abundance of Tbp (~23 Cts) used as endogenous control. We now provided also a number of new western blots measuring *Mettl3* protein expression in the various conditions used.

Figure 2A should be supplementary

We moved the panel as recommended.

Figure 2B, first two quadrants have no labels no RNP and Mettl3 RNP?

Thank you. We corrected this.

The differences described are relative mild, but the authors use language that is too strong such as “profound”, “crucial”. Should tone it down

We now revised our manuscript to improve the language.

Please revise the use of uppercase or lowercase letters and italics for Mettl3 throughout to use the proper designation (mouse vs human; gene vs protein).

We checked the text for consistent use of the designation.

References:

1. Patil, D.P., Pickering, B.F. & Jaffrey, S.R. Reading m(6)A in the Transcriptome: m(6)A-Binding Proteins. *Trends Cell Biol* **28**, 113-127 (2018).
2. Montagner, S. et al. TET2 Regulates Mast Cell Differentiation and Proliferation through Catalytic and Non-catalytic Activities. *Cell Rep* **15**, 1566-1579 (2016).
3. Su, S. et al. Cryo-EM structures of human m(6)A writer complexes. *Cell Res* **32**, 982-994 (2022).
4. Yan, X. et al. AI-empowered integrative structural characterization of m(6)A methyltransferase complex. *Cell Res* **32**, 1124-1127 (2022).
5. Geula, S. et al. Stem cells. m6A mRNA methylation facilitates resolution of naïve pluripotency toward differentiation. *Science* **347**, 1002-1006 (2015).

REVIEWERS' COMMENTS

Reviewer #3 (Remarks to the Author):

The authors made a considerable effort in addressing the reviewer's comments, although not all comments were fully addressed. For IL13 production by mast cells the authors tightened the concept that methylation by Mettl3 complexes in consensus m6A sites of the Il13 3'-untranslated region leads to reduced Il13 message stability. Additionally, they also provide support for a role of RNA methylation on mast cell responses by depleting other components of the methyl transferase complex such as Mettl14 with similar results.

For the other mast cell functions affected by Mettl3 knockdown, the mechanisms are still not clear. Although the authors provide new data on differentially expressed genes in Mettl3 knockdown cells, which in broad strokes may partly explain the other functional differences, the mechanisms for those changes in abundance of messages or the specific culprits are not defined. A detailed follow up on such questions may not be reasonable for this study, already heavy on data. However, the manuscript would benefit from some editing where the authors state more clearly throughout the paper that from all the affected mast cell functions, they focused on the mechanisms of cytokine regulation, in particular Il13, and the reasons for this focus.

In addition, I miss some analysis or discussion about how the RNASeq data for the knockdown of Mettl3 intersects/agrees with the nanostring panel using the other model of Mettl3 depletion. How many of the messages upregulated using the nanostring (such as those shown in Figure 7E) were also represented (upregulated) in the comparisons in Fig5? If IL13 is in that list it will also help focusing further on the mechanisms of IL13 regulation

Minor:

Please be consistent with the use of KO vs RNP in the figures, for example in Figure 5 it is KO, in others in Mettl3 RNP.

The abstract and intro talk about results after deletion of Mettl14 but Mettl4 should be introduced. For example, line 37 in the abstract (and also similarly in the intro): "Mechanistically, deletion of Mettl3 or Mettl14 led to the enhanced expression of inflammatory cytokines" should include something like "Mechanistically, deletion of Mettl3 or Mettl14, another component of the methyltransferase complex, led to the enhanced expression of inflammatory cytokines" or whatever the authors choose to define Mettl14.

Figure 1a- Are the labels of Unstimulated, IgE, IgE+DNP correct? Lane 2 is 0.5 h with IgE+DNP or is it IgE alone?

Figure 1B and Supplem 1D do not have labels for the curves

Did not find Supplementary Figure 1g referred to in the text. Please check

I did not find the supplementary Tables 4 and 5

In methods within the "mast cell activation description", the sensitization and challenge protocol should be described in detail as described in the degranulation section, while in the degranulation section, the description does not need to be detailed as much. Also, for the IL-33 treatment, were the cells preincubated with IL-33 or, alternatively, Ag and IL33 were added simultaneously? This has been shown to be consequential for the effects of IL33 on Antigen-induced responses

Reviewer #4 (Remarks to the Author):

Overall this is a very nice and professional manuscript that examines the effects of METTL3 in mast cells using sophisticated approaches for efficient KO in these difficult cell types. The clear description of the effects (and how they differ from other cell types) is good. I think the work is convincing.

I wish the authors had measured m6A levels after their manipulation of METTL3 (KD/KO and overexpression). But this was not requested by the reviewers. The authors should comment that m6A measurements would be useful to further confirm that the m6A levels are changing (and to what degree).

The authors refer to m6A-CLIP. The original name was miCLIP as used in ref. 41. I would call it miCLIP or if they wish m6A-CLIP-Seq/miCLIP and then use whichever acronym they like.

Just because a gene/mRNA contains m6A, it does not mean that m6A (or its loss) accounts for its upregulation in the KO cells. m6A needs to be present in high numbers and high stoichiometry. So I would temper the logical connection between the mapping data and the putative mechanism (i.e. m6A-mediated stability control). More studies such as stability measurements and mutation of the m6A site are needed and are beyond the scope.

When referring to the functions of YTHDF proteins - they should cite Zaccara et al, Cell 2020 which showed the redundant functions. Note the link of YTHDF to translation was shown not to be correct in this study, so the authors can say that the clearest function of YTHDF proteins is to mediate m6A-mediated mRNA instability in a redundant manner and cite the Zaccara paper.

Point-by-point reply to the Reviewers' comments:

We thank the Reviewers for providing constructive comments on our manuscript. A detailed response to their remaining observations is provided below.

Reviewer #3 (Remarks to the Author):

The authors made a considerable effort in addressing the reviewer's comments, although not all comments were fully addressed. For IL13 production by mast cells the authors tightened the concept that methylation by Mettl3 complexes in consensus m6A sites of the Il13 3'-untranslated region leads to reduced Il13 message stability. Additionally, they also provide support for a role of RNA methylation on mast cell responses by depleting other components of the methyl transferase complex such as Mettl14 with similar results.

For the other mast cell functions affected by Mettl3 knockdown, the mechanisms are still not clear. Although the authors provide new data on differentially expressed genes in Mettl3 knockdown cells, which in broad strokes may partly explain the other functional differences, the mechanisms for those changes in abundance of messages or the specific culprits are not defined. A detailed follow up on such questions may not be reasonable for this study, already heavy on data. However, the manuscript would benefit from some editing where the authors state more clearly throughout the paper that from all the affected mast cell functions, they focused on the mechanisms of cytokine regulation, in particular Il13, and the reasons for this focus.

We now improved the text as recommended, by stating the central importance of cytokines for the outcome of immune responses (thereby the rationale for focusing on them) and the reason behind our focus on IL-13 in the last few panels of the manuscript (it is the most prominently affected cytokine in all experimental conditions).

In addition, I miss some analysis or discussion about how the RNASeq data for the knockdown of Mettl3 intersects/agrees with the nanostring panel using the other model of Mettl3 depletion. How many of the messages upregulated using the nanostring (such as those shown in Figure 7E) were also represented (upregulated) in the comparisons in Fig5? If IL13 is in that list it will also help focusing further on the mechanisms of IL13 regulation

We thank the reviewer for pointing this out. We did not include this analysis in the revised manuscript because of the different conditions in which the two experiments were performed (unstimulated cells for RNA-seq and stimulated cells for Nanostring). Since the Nanostring panel can primarily detect inducible inflammatory transcripts, its overlap with unstimulated RNA-seq data is minimal, as expected in such experimental setup. We now improved the text to clarify this point.

Minor:

Please be consistent with the use of KO vs RNP in the figures, for example in Figure 5 it is KO, in others in Mettl3 RNP.

We thank the reviewer for pointing this out. We corrected the figure.

The abstract and intro talk about results after deletion of Mettl14 but Mettl4 should be introduced. For example, line 37 in the abstract (and also similarly in the intro): "Mechanistically, deletion of Mettl3 or Mettl14 led to the enhanced expression of inflammatory cytokines" should include something like

“Mechanistically, deletion of Mettl3 or Mettl14, another component of the methyltransferase complex, led to the enhanced expression of inflammatory cytokines” or whatever the authors choose to define Mettl14.

We thank the reviewer for pointing this out. We now revised the text as recommended.

Figure 1a- Are the labels of Unstimulated, IgE, IgE+DNP correct? Lane 2 is 0.5 h with IgE+DNP or is it IgE alone?

We thank the reviewer for pointing this out, lane 2 was indeed IgE alone. We corrected the figure.

Figure 1B and Supplem 1D do not have labels for the curves

We thank the reviewer for pointing this out. We corrected these figures.

Did not find Supplementary Figure 1g referred to in the text. Please check

Thank you, it is now properly included in the text.

I did not find the supplementary Tables 4 and 5

We now reorganized all the supplementary material as recommended by the editorial office.

In methods within the “mast cell activation description”, the sensitization and challenge protocol should be described in detail as described in the degranulation section, while in the degranulation section, the description does not need to be detailed as much. Also, for the IL-33 treatment, were the cells preincubated with IL-33 or, alternatively, Ag and IL33 were added simultaneously? This has been shown to be consequential for the effects of IL33 on Antigen-induced responses

We thank the reviewer for pointing this out. Antigen and IL-33 were added simultaneously. We now revised the text as recommended.

Reviewer #4 (Remarks to the Author):

Overall this is a very nice and professional manuscript that examines the effects of METTL3 in mast cells using sophisticated approaches for efficient KO in these difficult cell types. The clear description of the effects (and how they differ from other cell types) is good. I think the work is convincing.

I wish the authors had measured m6A levels after their manipulation of METTL3 (KD/KO and overexpression). But this was not requested by the reviewers. The authors should comment that m6A measurements would be useful to further confirm that the m6A levels are changing (and to what degree).

We thank the reviewer for pointing this out. We now revised the text in the Discussion to mention this limitation of our study.

The authors refer to m6A-CLIP. The original name was miCLIP as used in ref. 41. I would call it miCLIP or if they wish m6A-CLIP-Seq/miCLIP and then use whichever acronym they like.

We thank the reviewer for pointing this out. We now used the original name miCLIP throughout the text and figures.

Just because a gene/mRNA contains m6A, it does not mean that m6A (or its loss) accounts for its upregulation in the KO cells. m6A needs to be present in high numbers and high stoichiometry. So I would temper the logical connection between the mapping data and the putative mechanism (i.e. m6A-mediated stability control). More studies such as stability measurements and mutation of the m6A site are needed and are beyond the scope.

We agree with the reviewer. We were able to perform these experiments (stability measurement and mutations of the m⁶A sites) for the *Il13* transcript, but it is undoubtedly correct that the results cannot be generalized to all transcripts.

When referring to the functions of YTHDF proteins - they should cite Zaccara et al, Cell 2020 which showed the redundant functions. Note the link of YTHDF to translation was shown not to be correct in this study, so the authors can say that the clearest function of YTHDF proteins is to mediate m6A-mediated mRNA instability in a redundant manner and cite the Zaccara paper.

We modified the text to accommodate the requested citation.